# Error Propagation Mechanisms and Compensation Strategies for Quantized Diffusion Models

Songwei Liu [1] [*]   Chao Zeng [1] [*]   Chenqian Yan [1]   Xurui Peng [1]   Xing Wang [1]   Fangmin Chen [1]   Xing Mei [1]

## Abstract

Diffusion models have transformed image synthesis by establishing unprecedented quality and creativity benchmarks. Nevertheless, their large-scale deployment faces challenges due to computationally intensive iterative denoising processes. Although post-training quantization (PTQ) provides an effective pathway for accelerating sampling, the iterative nature of diffusion models causes stepwise quantization errors to accumulate progressively during generation, inevitably compromising output fidelity. To address this challenge, we develop a theoretical framework that mathematically formulates error propagation in Diffusion Models (DMs), deriving per-step quantization error propagation equations and establishing the first closed-form solution for cumulative error. Building on this theoretical foundation, we propose a timestep-aware cumulative error compensation scheme. Extensive experiments on multiple image datasets demonstrate that our compensation strategy effectively mitigates error propagation, significantly enhancing existing PTQ methods. Specifically, it achieves a 1.2 PSNR improvement over SVDQuant on SDXL W4A4, while incurring only an additional $< 0.5\%$ time overhead.

## 1. Introduction

DMs (Podell et al., 2023; Chen et al., 2024) have established themselves as highly effective deep generative frameworks across diverse domains, including image synthesis (Ho et al., 2020), video creation (Yang et al., 2024) and image translation (Sasaki et al., 2021) etc. Compared with conventional SOTA generative adversarial networks (GANs), DMs ex-

hibit superior stability, free from the common pitfalls of model collapse and posterior collapse, which ensures more reliable and diverse output generation. Although diffusion models demonstrate remarkable capabilities in generating high-fidelity and diverse images, their substantial computational and memory overhead hinders widespread adoption. This complexity primarily stems from two factors: first, the reliance on complex deep neural networks (DNNs) for noise estimation; second, the requirement for an iterative progressive denoising process to maintain synthesis quality, which may involve up to 1,000 iterative steps, substantially increasing computational demands.

To address the substantial computational demands of noise estimation in diffusion models, researchers employ quantization techniques to accelerate inference across all denoising steps. Depending on whether they require fine-tuning, quantization methods can be categorized into Quantization-Aware Training (QAT) (Wu et al., 2020) and PTQ (Cai et al., 2020). QAT necessitates retraining neural networks with simulated quantization and hyperparameter optimization, which introduces significant computational overhead and deployment complexity, making it unsuitable for compute-intensive diffusion model training. In contrast, PTQ directly derives quantization correction coefficients through calibration data statistics, thus avoiding the high-cost retraining process inherent to diffusion models. Although PTQ has been widely studied in traditional vision tasks such as image classification and object detection (Bhalgat et al., 2020), it faces many different challenges in diffusion models. The architectural characteristics and training strategies of diffusion models inherently lead to the widespread presence of outliers in weight distributions, while activation values exhibit step-wise distributional variations across time steps (Li et al., 2023a). These properties pose significant challenges for quantization by inducing substantial step-wise error. Furthermore, the iterative denoising mechanism amplifies error accumulation across successive steps, where quantization errors progressively accumulate during the sampling trajectory, ultimately degrading generation fidelity. Recent advancements in PTQ (Wu et al., 2024; Li et al., 2023a; Zhao et al., 2024b) have predominantly focused on minimizing quantization errors at individual denoising steps. However, these approaches systematically neglect the critical analy-

---

[*]Equal contribution [1]ByteDance Inc. Correspondence to: Fangmin Chen <cfangmin@gmail.com>, Xing Mei <xing.mei@bytedance.com>.

*Proceedings of the 43rd International Conference on Machine Learning*, Seoul, South Korea. PMLR 306, 2026. Copyright 2026 by the author(s).

sis of error propagation dynamics throughout the iterative sampling trajectory. Consequently, current solutions (Li et al., 2024b) remain constrained to 4-bit quantization of both weights and activations while maintaining acceptable quality degradation, as cumulative errors across sequential denoising stages fundamentally limit lower-bit quantization viability.

Contrarily, our work focuses on the quantization error propagation problem in diffusion models and proposes the first **t**imestep-aware **c**umulative **e**rror **c**ompensation scheme, called **TCEC**. First, we construct an error propagation equation by taking the DDIM (Song et al., 2020) as a paradigmatic case, presenting the field's inaugural closed-form solution for cumulative error. However, direct computation of cumulative errors proves computationally infeasible. Subsequently, through the implementation of timestep-sensitive online rapid estimation for per-step quantization error, we achieve a notable simplification of the computational complexity inherent in cumulative error modeling. Finally, as shown in Figure 1, we incorporate a cumulative error correction term in each generation step, dynamically mitigating errors induced by quantization. In summary, our contributions are:

- We experimentally demonstrate that cumulative error is the primary cause of poor performance in low-precision DMs, thus presenting TCEC in response. Unlike conventional PTQ methods, TCEC directly computes cumulative errors and integrates correction terms during the iterative sampling process to align the outputs of quantized models with their floating-point counterparts.

- To accurately compute cumulative errors, we present a theoretical framework encompassing three key components: per-step quantization error, cumulative error, and error propagation. For the first time in the field, we derive a closed-form solution for cumulative error. Through rational approximations, we substantially simplify its computational complexity, enabling low-cost and rapid correction of cumulative errors.

- Extensive experiments across various diffusion models demonstrate our method's effectiveness. TCEC-W4A4 reduces the memory footprint by 3.5× compared to the FP16 model and accelerates inference by 3× versus NF4 weight-only quantization, with engineering performance comparable to SVDQuant (Li et al., 2024b). Across all precision levels, it achieves superior image fidelity and diversity—for example, an sDCI PSNR↑ of 21.9 (vs. 20.7) and an MJHQ FID↓ of 18.1 (vs. 20.6).

Notably, TCEC maintains orthogonality to existing state-of-the-art PTQ algorithms (Li et al., 2024b; Zhao et al., 2024a)

that minimize per-step quantization errors. Additionally, TCEC originates from rigorous theoretical derivation, and while its derivation process uses DDIM as an example, it is equally applicable to other solvers such as DPM++ (Lu et al., 2022).

## 2. Related Work

### 2.1. Diffusion Model

Diffusion models are a family of probabilistic generative models that progressively destruct real data by injecting noise, then learn to reverse this process for generation, represented notably by denosing diffusion probabilistic models (DDPMs) (Ho et al., 2020). DDPM is composed of two Markov chains of $T$ steps. One is the forward process, which incrementally adds Gaussian noises into real sample $x_0 \in q(x_0)$, In this process, a sequence of latent variables $x_{1:T} = [x_1, x_2, ..., x_T]$ are generated in order and the last one $x_T$ will approximately follow a standard Gaussian:

$$q(\mathbf{x}_t \mid \mathbf{x}_{t-1}) = \mathcal{N}(\mathbf{x}_t; \sqrt{1 - \beta_t}\mathbf{x}_{t-1}, \beta_t\mathbf{I}) \qquad (1)$$

where $\beta_t$ is the variance schedule that controls the strength of the Gaussian noise in each step. The reverse process removes noise from a sample from the Gaussian noise input $\mathbf{x}_T \sim \mathcal{N}(\mathbf{0}, \mathbf{I})$ to gradually generate high-fidelity images. However, since the real reverse conditional distribution $q(x_{t-1}|x_t)$ is unavailable, diffusion models sample from a learned conditional distribution:

$$p_\theta(\mathbf{x}_{t-1} \mid \mathbf{x}_t) = \mathcal{N}\left(\mathbf{x}_{t-1}; \mu_\theta(\mathbf{x}_t, t), \Sigma_\theta(\mathbf{x}_t, t)\right) \quad (2)$$

where $p(\mathbf{x}_T) \sim \mathcal{N}(\mathbf{x}_T; \mathbf{0}, \mathbf{I})$, with $\mu_\theta(\mathbf{x}_t, t)$ denotes the noise estimation model, and $\Sigma_\theta(\mathbf{x}_t, t)$ denotes the variance for sampling which can be fixed to constants (Luo, 2022). The denoising process, constrained by the Markov chain, requires a huge number of iterative time steps in DDPM. DDIM generalizes the diffusion process to non-Markovian processes, simulating the diffusion process with fewer steps. It has replaced DDPM as the mainstream inference strategy. Our work focuses on accelerating the inference of the noise estimation model in DDIM, with a training-free PTQ process.

### 2.2. Post-Training Quantization

The noise estimation models such as UNET (Podell et al., 2023) or Transformer (Chen et al., 2024; Yang et al., 2024; Zheng et al., 2024) exhibit high computational complexity, rendering the sampling of diffusion models computationally expensive. PTQ transforms the weights and activations of the full-precision model into a low-bit format, enabling the model's inference process to utilize the integer matrix multiplication units on the target hardware platform and accelerating the computational process (Jacob et al., 2018). Prior

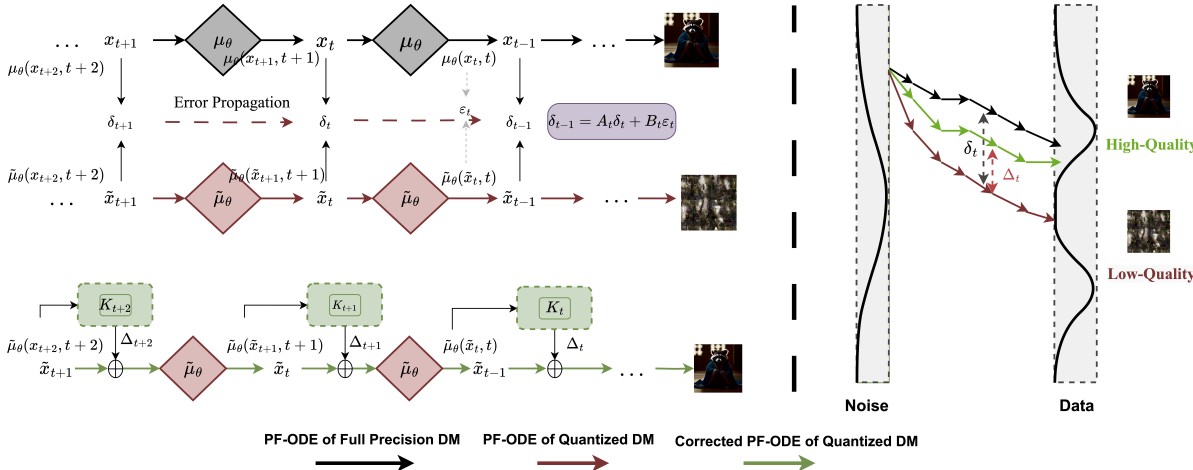

*Figure 1.* Visualization of Error Propagation and Correction in Quantized Diffusion Models. The gray path represents the iterative denoising process of the full-precision model $\mu$, while the brown-red path represents that of the quantized model $\tilde{\mu}$. Affected by cumulative errors $\delta_t$, its output gradually deviates from $\mu$. The green path represents the denoising process after online correction of cumulative errors, with outputs better aligned with the full-precision model.

studies, such as PTQD (He et al., 2023) and Q-DM (Li et al., 2023b), have explored the application of quantization techniques for diffusion models. Q-Diffusion (Li et al., 2023a) and PTQ4DM (Shang et al., 2023) first achieved 8-bit quantization in text-to-image generation tasks. Subsequent research refined these methodologies through strategies such as sensitivity analysis (Yang et al., 2023) and timestep-aware quantization (Huang et al., 2024a; Wang et al., 2023). Among these works, MixDQ (Zhao et al., 2024b) introduces metric-decoupled sensitivity analysis and develops an integer programming-based method to derive optimal mixed-precision configurations. Qua2SeDiMo (Mills et al., 2025) enables high-quality mixed-precision quantization decisions for a wide range of diffusion models, from foundational U-Nets to state-of-the-art Transformers, extending the quantization lower bounds for image generation tasks to W4A8. SVDQuant (Li et al., 2024b) enhances quantization performance by integrating fine-grained quantization with singular value decomposition (SVD)-based weight decomposition, achieving W4A4 quantization while maintaining acceptable quality degradation. Meanwhile, ViDiT-Q (Zhao et al., 2024a) further explores quantization for video generation tasks (Yang et al., 2024; Zheng et al., 2024), achieving W8A8 and W4A8 with negligible degradation in visual quality and metrics. These works minimize per-step errors through more precise quantization approximations at the layer level, yet overlook the error propagation in the diffusion process. Particularly in video generation tasks, which require a greater number of inference steps, the issue of error propagation becomes exacerbated, and accumulated errors across sequential denoising stages fundamentally constrain the feasibility of lower-bit quantization. Although several studies (Chu et al., 2024; Yao et al., 2024) have

recognized the issue of error propagation and attempted to propose solutions, their efforts are focused on "single-step error source suppression" (e.g., TAC decomposes input/noise errors, while QNCD targets noise in the embedding layer). These works neither elucidate the relationship between single-step errors and cumulative errors nor validate the model performance beyond small academic datasets such as CIFAR10.

Unlike prior studies, our method TCEC focuses on the error propagation problem in quantized diffusion models. We develop a theoretical framework that, through rigorous analysis, first models the relationship between per-step quantization errors and cumulative errors, derives a closed-form solution for cumulative errors, and then provides strategies to dynamically mitigate error accumulation during each denoising step.

## 3. Method

In this section, we first formally formulate the error propagation dynamics in quantized diffusion models and develop a preliminary analytical solution. Next, we establish theoretically-grounded approximations to reduce the computational complexity associated with cumulative error tracking. Finally, we propose a timestep-aware online estimation framework for per-step quantization errors, culminating in the TCEC mechanism - an efficient solution for real-time cumulative error mitigation.

### 3.1. Error Propagation Mechanisms

The iterative denoising process of diffusion models corresponds to the discrete approximation of the probability

FP16     GGUF W4     ViDiT-Q W4A4     SVDQuant W4A4     TCEC W4A4

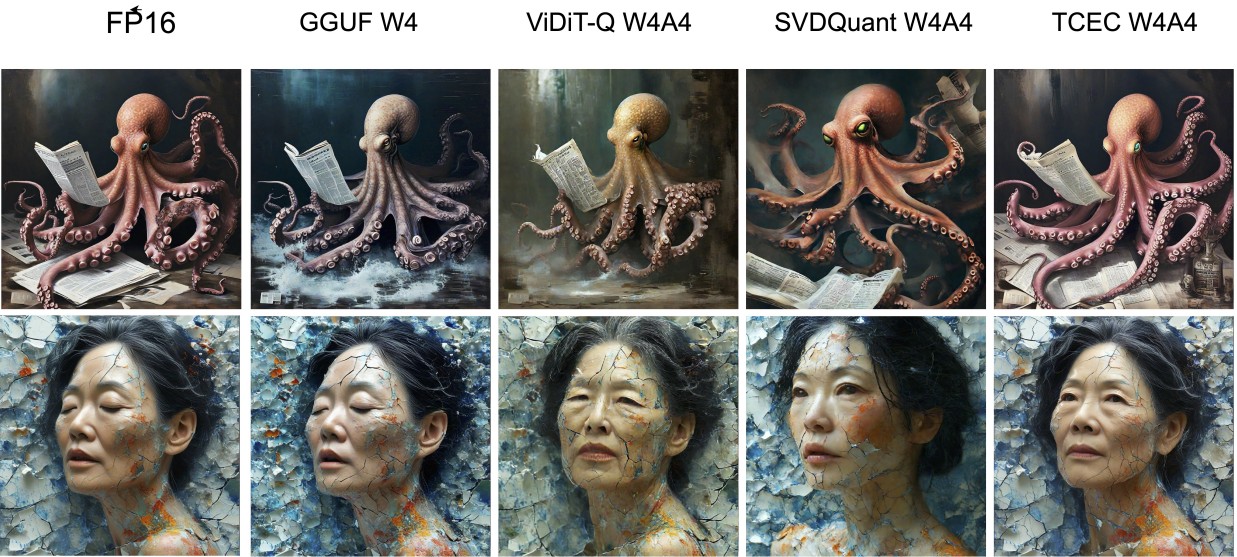

*Figure 2.* Qualitative visual results comparison. **Prompt1**: *An alien octopus floats through a portal reading a newspaper.* **Prompt2**: *A middle-aged woman of Asian descent, her dark hair streaked with silver, appears fractured and splintered, intricately embedded within a sea of broken porcelain. The porcelain glistens with splatter paint patterns in a harmonious blend of glossy and matte blues, greens, oranges, and reds, capturing her dance in a surreal juxtaposition of movement and stillness. Her skin tone, a light hue like the porcelain, adds an almost mystical quality to her form.*

flow ordinary differential equation(PF-ODE), the noise at each time step $t \in [T, ..., 1]$ is computed from $x_t$ by a full-precision noise estimation model $\mu_\theta$ whose weights are fixed at all steps. Based on DDIM-solver, we can calculate the sample $\mathbf{x}_{t-1}$ at time $t-1$ as follows:

$$x_{t-1} = \frac{\sqrt{\alpha_{t-1}}}{\sqrt{\alpha_t}} x_t$$
$$+ \left( \sqrt{1 - \alpha_{t-1}} - \frac{\sqrt{\alpha_{t-1}(1 - \alpha_t)}}{\sqrt{\alpha_t}} \right) \mu_\theta(x_t, t) \tag{3}$$

where $\alpha_t$ is a constant related to the noise schedule $\beta_t$ and the specific relationship is $\frac{\alpha_t}{\alpha_{t-1}} = 1 - \beta_t$. Since $\beta_t \in (0, 1)$, $\alpha_t$ is monotonically decreasing with respect to $t$. We denote the quantized version of the noise estimation model as $\tilde{\mu}_\theta$. When the input remains unchanged, it is formulated by:

$$\tilde{\mu}_\theta(\tilde{x}_t, t) = \mu_\theta(\tilde{x}_t, t) + \varepsilon_t \tag{4}$$

where $\varepsilon_t$ means per-step quantization error which is introduced due to model quantization and only relies on the module at iteration $t$ and is independent of the others. The amount of error accumulated by continuously running the first $T - t + 1$ denoising steps is called the cumulative error $\delta_t$, then the input including cumulative error can be expressed as $\tilde{x}_t = x_t + \delta_t$. Consequently, referring to Eq. 3, the iterative process of the quantized model can be expressed as:

$$\tilde{x}_{t-1} = \frac{\sqrt{\alpha_{t-1}}}{\sqrt{\alpha_t}} \tilde{x}_t$$
$$+ \left( \sqrt{1 - \alpha_{t-1}} - \frac{\sqrt{\alpha_{t-1}(1 - \alpha_t)}}{\sqrt{\alpha_t}} \right) \tilde{\mu}_\theta(\tilde{x}_t, t) \tag{5}$$

Based on Eq. 4 and the definition of cumulative error, Eq. 5 can be reformulated as:

$$x_{t-1} + \delta_{t-1} = \frac{\sqrt{\alpha_{t-1}}}{\sqrt{\alpha_t}} (x_t + \delta_t)$$
$$+ \left( \sqrt{1 - \alpha_{t-1}} - \frac{\sqrt{\alpha_{t-1}(1 - \alpha_t)}}{\sqrt{\alpha_t}} \right) \tag{6}$$
$$\cdot \left( \mu_\theta(x_t + \delta_t, t) + \varepsilon_t \right)$$

Applying the first-order Taylor expansion, $\mu_\theta(x_t + \delta_t, t)$ is approximated as $\mu_\theta(x_t, t) + \mathbf{J}_{x_t} \delta_t$. Substituting it into Eq. 6, we can obtain the error propagation equation that relates the per-step quantization error to the cumulative error:

$$\delta_{t-1} = A_t \delta_t + B_t \varepsilon_t \tag{7}$$

in which $A_t = \frac{\sqrt{\alpha_{t-1}}}{\sqrt{\alpha_t}} I + B_t * \mathbf{J}_{x_t}$, $B_t = \sqrt{1 - \alpha_{t-1}} - \frac{\sqrt{\alpha_{t-1}(1 - \alpha_t)}}{\sqrt{\alpha_t}}$ and $J_{x_t} = \nabla_{x_t} \mu_\theta(x_t, t)$ is the Jacobian matrix of the denoising model $\mu_\theta$. Given that $\delta_T = 0$, when

we recursively expand Eq. 7 from $T$ to $t$, the cumulative error $\delta_t$ can be derived as:

$$\delta_t = \sum_{k=t}^{T} \left( \prod_{j=t}^{k-1} A_j^{-1} \right) B_k \varepsilon_k \qquad (8)$$

With this equation, we obtain a closed-form solution for the cumulative error corresponding to step $t$. By directly adding a correction term of $\Delta_t = -\delta_t$ to Eq. 3, the error introduced by quantization can be rectified. However, there are two issues in directly calculating Eq. 8: **Issue 1**—the existence of superimposed continuous multiplication and addition and the second derivative of the matrix makes the computational complexity unacceptable, and **Issue 2**—there is no explicit analytical solution for the per-step quantization error.

### 3.2. Simplify Computational Complexity

In this section, we tackle **Issue 1** via reasonable approximation, thus simplifying the computational complexity of $\Delta_t$.

**Approximation 1** *For a well-trained diffusion model, it is insensitive to local changes in the input, which implies that we can ignore the Jacobian term: $J_{x_t} \approx 0$. See Appendix A for details.*

Consequently, the inverse of the propagation matrix $\mathbf{A}_j$ can be approximately represented as $\frac{\sqrt{\alpha_j}}{\sqrt{\alpha_{j-1}}}\mathbf{I}$. By expanding the product terms in Eq. 8 and reducing the intermediate terms, the final result can be expressed as:

$$\prod_{j=t}^{k-1} A_j^{-1} = \prod_{j=t}^{k-1} \frac{\sqrt{\alpha_j}}{\sqrt{\alpha_{j-1}}} = \frac{\sqrt{\alpha_t}}{\sqrt{\alpha_{t-1}}} \cdot \frac{\sqrt{\alpha_{t+1}}}{\sqrt{\alpha_t}} \cdots = \frac{\sqrt{\alpha_{k-1}}}{\sqrt{\alpha_{t-1}}} \qquad (9)$$

Plugging Eq. 9 into Eq. 8, we have:

$$\Delta_t = -\sum_{k=t}^{T} \left( \frac{\sqrt{\alpha_{k-1}}}{\sqrt{\alpha_{t-1}}} \right) B_k \varepsilon_k \qquad (10)$$

Although the solution form has been greatly simplified, there are still difficulties, such as high computational complexity from time step $T$ to $t$. To further speed up computation, we introduce an additional approximation, whose rationality will be rigorously proven later.

**Approximation 2** *The correction term only takes into account the subsequent $m$ steps. Since the denoising process unfolds in reverse, proceeding from $T$ to $0$, at the $t-th$ step, only the quantization noises at steps $t + m$, $t + m - 1, \cdots, t+1$ are factored in. We refer to this as the temporal locality approximation.*

Therefore, we can effectively reformulate Eq. 10 as:

$$\Delta_t \approx -\frac{1}{\sqrt{\alpha_{t-1}}} \sum_{k=t}^{\min(t+m,T)} \sqrt{\alpha_{k-1}}\mathbf{B}_k \varepsilon_k \qquad (11)$$

**How to determine the value of the parameter $m$?** Based on our discussion in Sec. 3.1, the cumulative error at step $t - 1$ is related to the cumulative error $\delta_t$ from steps $[T, \ldots, t+1]$ and the per-step quantization error $\varepsilon_t$ at step $t$. By substituting the correction term $\Delta_t$ into Eq. 7, we have:

$$\widehat{\delta_{t-1}} = \mathbf{A}_t \delta_t + \mathbf{B}_t \varepsilon_t + \Delta_t$$
$$= A_t \delta_t - \frac{1}{\sqrt{\alpha_{t-1}}} \sum_{k=t+1}^{\min(t+m,T)} \sqrt{\alpha_{k-1}} B_k \varepsilon_k \qquad (12)$$

where $\widehat{\delta_{t-1}}$ is the corrected cumulative error and it should exhibits a strictly smaller upper bound in norm compared to $\delta_{t-1}$, signifying a more refined and accurate error representation. Based on this, we can solve for the reasonable value of $m$. Under mild regularity conditions, there exists $\sigma > 0$ independent of the timestep $k$, such that the per-step quantization error satisfies $\|\varepsilon_k\| \leq \sigma, \forall k$. Based on this condition, we can deduce from $\|\widehat{\delta_0}\| \leq \sigma \sum_{t=1}^{T} \left( \prod_{k=1}^{t-1} \rho_k \right) \|\mathbf{B}_t\| \leq \|\delta_0\|$ that $m = 1$. We provide a complete proof to this theorem in Appendix B and an empirical study demonstration in Appendix C. This implies that the cumulative error at any timestep is only related to the per-step quantization errors of the two immediately preceding steps. Thus, the final cumulative error correction term can be reformulated as:

$$\Delta_t \approx -\frac{1}{\sqrt{\alpha_{t-1}}} \sum_{k=t}^{\min(t+1,T)} \sqrt{\alpha_{k-1}}\mathbf{B}_k \varepsilon_k \qquad (13)$$

### 3.3. Timestep-Aware Compensation

In this section, we address **Issue 2** to derive the definitive form of the correction terms. By visualizing and analyzing the noise estimation of the full-precision model ($\mu_\theta(\tilde{x}_t, t)$), the noise estimation of the quantized model ($\tilde{\mu}_\theta(\tilde{x}_t, t)$), and the output distortion ($\varepsilon_t$) across different time steps in Appendix E, two critical empirical observations emerge:**Timestep-Dependent Error Characteristics**—the per-step quantization error $\varepsilon_t$ exhibits significant variations across timesteps, with distinct spatial and magnitude patterns at different stages of the denoising process. **Output-Correlated Error Propagation**—a strong statistical correlation exists between $\varepsilon_t$ and the quantized model's outputs $\tilde{\mu}_\theta(\tilde{x}_t, t)$, particularly in high-frequency regions.

These findings motivate our core proposition: *The per-step quantization error $\varepsilon_t$ can be partially reconstructed by adaptively scaling the quantized noise estimates $\tilde{\mu}_\theta(\tilde{x}_t, t)$ with channel-specific coefficients.* Formally, we define the per-step quantization error as:

$$\varepsilon_t = \mathbf{K}_t \odot \tilde{\mu}_\theta(\tilde{x}_t, t) \qquad (14)$$

where $\mathbf{K}_t \in \mathbb{R}^C$ denotes a timestep-conditioned channel-wise scaling matrix, and $\odot$ represents element-wise multiplication. We now focus on finding a loss function $L$, by minimizing which, we can efficiently reconstruct $\mu_\theta(\tilde{x}_t, t)$, $\forall t \in [0, T]$. We adopt the Mean Squared Error (MSE) as the loss function and introduce regularization terms to prevent overfitting :

$$\mathcal{L}(\mathbf{K}) = \sum_{t=1}^{T} \sum_{i=1}^{C} \sum_{j=1}^{H} \sum_{k=1}^{W} [(1 - K_{t,i})\tilde{\mu}_{t,i,j,k} - \mu_{t,i,j,k}]^2$$
$$+ \lambda_1 \sum_{t=1}^{T} \sum_{i=1}^{C} K_{t,i}^2$$
(15)

where $T$ is the total denoising timesteps, $C$ is the number of noise estimation channels, $H \times W$ represents the spatial dimensions of feature maps and $\lambda_1$ restricts the magnitude of these coefficients. The loss function $\mathcal{L}(\mathbf{K})$ is strictly convex with respect to $K$, when $\lambda_1 > 0$. To derive the optimal scaling coefficients, set the first derivative to zero:

$$\frac{\partial \mathcal{L}}{\partial K_{t,i}} = -2 \sum_{j=1}^{H} \sum_{k=1}^{W} [(1 - K_{t,i})\tilde{\mu}_{t,i,j,k} - \mu_{t,i,j,k}]\tilde{\mu}_{t,i,j,k}$$
$$+ 2\lambda_1 K_{t,i} = 0$$
(16)

Rearranging the terms, we obtain:

$$K_{t,i} = \frac{\sum_{j=1}^{H} \sum_{k=1}^{W} \left( \tilde{\mu}_{t,i,j,k}^2 - \mu_{t,i,j,k}\tilde{\mu}_{t,i,j,k} \right)}{\sum_{j=1}^{H} \sum_{k=1}^{W} \tilde{\mu}_{t,i,j,k}^2 + \lambda_1} \quad (17)$$

To prevent division by zero and ensure a stable and efficient reconstruction of quantization errors while maintaining theoretical rigor, we add $\gamma = 10^{-8}$ to the denominator.

Based on the provided calibration data, We can cache the noise prediction outputs of the full-precision model and the quantized model, and then compute $K \in \mathbb{R}^{T \times C}$ offline, which is directly used for quantization errors reconstruction during inference. Different values of $\lambda_1$ correspond to different values of $K$. We offer two methods, grid search and empirical rule, to determine the optimal value and compare their advantages and disadvantages. According to the experiments in Appendix F, the latter is adopted, that means $\lambda_1 = 0.01 \times \frac{\text{mean}(\tilde{\mu}^2)}{\text{var}(\mu)}$. Substituting Eq. 14 into Eq. 13, we obtain the final form of the closed-form solution for cumulative error:

$$\Delta_t \approx -\frac{1}{\sqrt{\alpha_{t-1}}} \sum_{k=t}^{\min(t+1,T)} \sqrt{\alpha_{k-1}} \mathbf{B}_k \mathbf{K}_k \tilde{\mu}_\theta(\tilde{x}_k, k) \quad (18)$$

Eq. 18 indicates that relying on the outputs of the two immediately preceding steps $(t+1, t))$ of the quantized diffusion

model, one can achieve a rapid estimation of the cumulative error at the current step $(t)$. This estimation stems from strict theoretical derivations, with the extra cost entailing minimal computations and caching the output of step $(t + 1)$, which usually involves feature maps in a compact latent space.

## 4. Experiments

### 4.1. Implementation Details

**Quantization Scheme.** SVDQuant (Li et al., 2024b) introduces an additional low-rank branch that can mitigate quantization challenges in both weights and activations, establishing itself as a new benchmark for PTQ algorithms. In this study, we build our quantization strategy upon it by integrating cumulative error correction mechanisms. This approach ensures that performance comparisons remain unaffected by operator-level quantization configurations. In the 8-bit configuration, our approach employs per-token dynamic quantization for activations and per-channel weight quantization, complemented by a low-rank auxiliary branch with a rank of 16. For the 4-bit configuration, we apply per-group symmetric quantization to both activations and weights, using a low-rank branch with rank 32 and setting the group size to 64. All nonlinear activation and normalization layers are not quantized, meanwhile the inputs of linear layers in adaptive normalization are kept in 16 bits. To comprehensively evaluate the effectiveness of TCEC, we conducted comparative experiments with recent SOTA quantization algorithms across diverse generation tasks.

**Image Generation Evaluation.** We benchmark TCEC using SDXL (Podell et al., 2023), SDXL-Turbo (Podell et al., 2023) and PixArt models (Chen et al., 2023; 2024) including both the UNet and DiT backbones. SDXL-Turbo uses the default configuration of 4 inference steps, while SDXL employs the DDIM sampler with 50 steps. Since PixArt utilizes the DPM++ solver, we adapted TCEC to this advanced high-order solver to demonstrate its compatibility across different solvers. The detailed validation process is presented in Appendix D. To precompute the channel-wise scaling matrices $K$ and $\lambda_1$ mentioned in section 3.3, we randomly sampled 1,024 prompts from COCO dataset (Chen et al., 2015) as the calibration dataset. Appendix I provides a detailed explanation of the selection and size of the calibration dataset. To evaluate the generalization capability of TCEC, we sample 5K prompts from the MJHQ-30K (Li et al., 2024a) and the summarized Densely Captioned Images(sDCI) (Urbanek et al., 2024) for benchmarking.

**Video Generation Evaluation.** We apply TCEC to Open-SORA (Zheng et al., 2024), the videos are generated with 100-steps DDIM with CFG scale of 4.0. We evaluate the quantized model on VBench (Huang et al., 2024b) to provide comprehensive results. Following prior research (Ren

*Table 1.* Quantization Performance Comparison of Different Models. SDXL and SDXL-Turbo generate at $512^2$ resolution, while PixArt achieves $1024^2$. Evaluation metrics include FID (distribution distance) (Parmar et al., 2024), IR (human preference) (Xu et al., 2023), LPIPS (perceptual similarity) (Zhang et al., 2018), and PSNR (numerical fidelity against 16-bit references) (Li et al., 2024b).

| Model | Precision | Method | MJHQ | | | | sDCI | | | |
|---|---|---|---|---|---|---|---|---|---|---|
| | | | Quality | | Similarity | | Quality | | Similarity | |
| | | | FID↓ | IR↑ | LPIPS↓ | PSNR↑ | FID↓ | IR↑ | LPIPS↓ | PSNR↑ |
| SDXL | FP16 | - | 16.6 | 0.729 | - | - | 22.5 | 0.573 | - | - |
| | W8A8 | TensorRT | 20.2 | 0.591 | 0.247 | 22.0 | 25.4 | 0.453 | 0.265 | 21.7 |
| | W8A8 | SVDQuant | 16.6 | 0.718 | 0.119 | 26.4 | 22.4 | 0.574 | 0.129 | 25.9 |
| | W8A8 | SVDQuant + TCEC | **16.0** | **0.728** | **0.092** | **27.3** | **22.0** | **0.580** | **0.103** | **26.7** |
| | W4A4 | SVDQuant | 20.6 | 0.601 | 0.288 | 21.0 | 26.3 | 0.477 | 0.307 | 20.7 |
| | W4A4 | SVDQuant + TCEC | **18.1** | **0.652** | **0.249** | **21.9** | **23.4** | **0.513** | **0.259** | **21.9** |
| SDXL-Turbo | FP16 | - | 24.3 | 0.845 | | | 24.7 | 0.705 | - | - |
| | W8A8 | MixDQ | **24.1** | 0.834 | 0.147 | 21.7 | 25.0 | 0.690 | 0.157 | 21.6 |
| | W8A8 | SVDQuant | 24.3 | 0.845 | 0.100 | 24.0 | 24.8 | 0.701 | 0.110 | 23.7 |
| | W8A8 | SVDQuant + TCEC | 24.5 | **0.849** | **0.083** | **24.9** | **23.9** | **0.720** | **0.098** | **24.5** |
| | W4A8 | MixDQ | 27.7 | 0.708 | 0.402 | 15.7 | 25.9 | 0.610 | 0.415 | 15.7 |
| | W4A4 | MixDQ | 353 | -2.26 | 0.685 | 11.0 | 373 | -2.28 | 0.686 | 11.3 |
| | W4A4 | SVDQuant | 24.6 | 0.816 | 0.262 | 18.1 | 26.0 | 0.671 | 0.272 | 18.0 |
| | W4A4 | SVDQuant + TCEC | **23.9** | **0.833** | **0.230** | **19.0** | **25.1** | **0.691** | **0.232** | **19.3** |
| PixArt-$\sum$ | FP16 | - | 16.6 | 0.944 | - | - | 24.8 | 0.966 | - | - |
| | W8A8 | ViDiT-Q | **15.7** | 0.944 | 0.137 | 22.5 | 23.5 | **0.974** | 0.163 | 20.4 |
| | W8A8 | SVDQuant | 16.3 | 0.955 | 0.109 | 23.7 | 24.2 | 0.969 | 0.129 | 21.8 |
| | W8A8 | SVDQuant + TCEC | 16.2 | **0.964** | **0.098** | **24.5** | 23.4 | 0.952 | **0.118** | **22.6** |
| | W4A4 | ViDiT-Q | 412 | -2.27 | 0.854 | 6.44 | 425 | -2.28 | 0.838 | 6.70 |
| | W4A4 | SVDQuant | 19.2 | 0.878 | 0.323 | 17.6 | 25.9 | 0.918 | 0.352 | 16.5 |
| | W4A4 | SVDQuant + TCEC | **18.1** | **0.903** | **0.285** | **18.3** | 25.3 | **0.934** | **0.304** | **16.9** |

et al., 2024), we evaluate video quality from three distinct dimensions using eight carefully selected metrics. *Aesthetic Quality* and *Imaging Quality* focus on assessing the quality of individual frames, independent of temporal factors. *Subject Consistency*, *Background Consistency*, *Motion Smoothness*, and *Dynamic Degree* measure cross-frame temporal coherence and dynamics. Finally, *Scene Consistency* and *Overall Consistency* gauge the alignment of the video with the user-provided text prompt. We collected 128 samples from WebVid (Nan et al., 2024) as the calibration dataset to calculate $K$ and $\lambda_1$.

### 4.2. Main Results

**Image Generation Evaluation.** As shown in Table 1, we conduct extensive experiments at two quantization precisions: W8A8 and W4A4. We observe that, across all precision levels, TCEC achieves better image fidelity and diversity, and it even matches the 16-bit results under W8A8 quantization. For UNet-based models, on SDXL, our W4A4 model substantially outperforms SVDQuant W4A4, the current SOTA 4-bit approach, achieving an sDCI PSNR of 21.9. This even surpasses TensorRT's W8A8 result of 21.7, demonstrating robust performance under lower-bit quantization. On SDXL-Turbo, MixDQ W4A4 exhibits abnormal FID and IR metrics, indicating quantization failure. This

highlights the greater difficulty of quantizing models with a small number of inference steps. Our W4A4 model surpasses SVDQuant by 0.017 and 0.02 in MJHQ IR and sDCI IR metrics, respectively, suggesting a stronger alignment with human preferences. The larger performance gains of TCEC on SDXL compared to SDXL-Turbo highlight the critical role of cumulative error correction in longer inference sequences, and we provide a complete exploration of the minimal inference steps required for high-speed inference in Appendix G. For DiT-based model, on PixArt-$\sum$, our W4A4 model significantly surpasses SVDQuant's W4A4 results across all metrics. Leveraging the DPM++ solver, PixArt-$\sum$ demonstrates TCEC's robustness across different solver configurations. As shown in Figure 2, when compared with ViDiT-Q W4A4 and SVDQuant W4A4, the TECE W4A4 method demonstrates less quality degradation and smaller changes in image content. More visual results can be found in Appendix K.

**Video Generation Evaluation.** As shown in Table 2, our evaluation compares SOTA PTQ methods under both W8A8 and W4A8 configurations. In Imaging Quality, TCEC achieves 65.56 (W8A8) and 64.97 (W4A8), surpassing ViDiT-Q by +2.08 and +3.90 absolute points respectively. It demonstrates significant advantages in frame-wise quality metrics that evaluate static visual fidelity independent of

*Table 2.* Performance of PTQ Algorithms for OpenSora on Vbench eval Benchmark in the W4A8 configuration.

| Method | Bit-width W/A | Imaging Quality | Aesthetic Quality | Motion Smooth. | Dynamic Degree | BG. Consist. | Subject Consist. | Scene Consist. | Overall Consist. |
|---|---|---|---|---|---|---|---|---|---|
| - | 16/16 | 63.68 | 57.12 | 96.28 | 56.94 | 96.13 | 90.28 | 39.61 | 26.21 |
| Q-Diffusion | 8/8 | 60.38 | 55.15 | 94.44 | 68.05 | 94.17 | 87.74 | 36.62 | 25.66 |
| Q-DiT | 8/8 | 60.35 | 55.80 | 93.64 | 68.05 | 94.70 | 86.94 | 32.34 | 26.09 |
| PTQ4DiT | 8/8 | 56.88 | 55.53 | 95.89 | 63.88 | 96.02 | 91.26 | 34.52 | 25.32 |
| SmoothQuant | 8/8 | 62.22 | 55.90 | 95.96 | **68.05** | 94.17 | 87.71 | 36.66 | 25.66 |
| Quarot | 8/8 | 60.14 | 53.21 | 94.98 | 66.21 | 95.03 | 85.35 | 35.65 | 25.43 |
| ViDiT-Q | 8/8 | 63.48 | 56.95 | 96.14 | 61.11 | 95.84 | 90.24 | 38.22 | 26.06 |
| ViDiT-Q + TCEC | 8/8 | **65.56** | **57.12** | **96.27** | 61.09 | **96.23** | **91.34** | **39.58** | **26.20** |
| Q-DiT | 4/8 | 23.30 | 29.61 | **97.89** | 4.166 | 97.02 | 91.51 | 0.00 | 4.985 |
| PTQ4DiT | 4/8 | 37.97 | 31.15 | 92.56 | 9.722 | **98.18** | **93.59** | 3.561 | 11.46 |
| SmoothQuant | 4/8 | 46.98 | 44.38 | 94.59 | 21.67 | 94.36 | 82.79 | 26.41 | 18.25 |
| Quarot | 4/8 | 44.25 | 43.78 | 92.57 | **66.21** | 94.25 | 84.55 | 28.43 | 18.43 |
| ViDiT-Q | 4/8 | 61.07 | 55.37 | 95.69 | 58.33 | 95.23 | 88.72 | 36.19 | 25.94 |
| ViDiT-Q + TCEC | 4/8 | **64.97** | **56.90** | 96.01 | 59.42 | 97.01 | 90.05 | **37.20** | **26.20** |

*Table 3.* Inference overhead comparison with TCEC. Quantization: video models use W8A8, Flux-dev uses W4A4.

| Model | Type | FP16 (s) | Quantized Inference (s) | |
|---|---|---|---|---|
| | | | w/o TCEC | w/ TCEC |
| Opensora1.2 (51 frames, 480P) | Video (W8A8) | 44.56 | 26.211 | **26.316** |
| CogVideoX (48 frames, 480P) | Video (W8A8) | 78.48 | 49.67 | **49.894** |
| Wan2.1-1.3B (81 frames, 480P) | Video (W8A8) | 199 | 118.45 | **119.029** |
| Flux-dev 1.0 (T = 30) | Img2Vid (W4A4) | 26.14 | 9.947 | **9.996** |

temporal factors. TCEC achieves 96.27 motion smoothness at 8-bit and 96.01 at 4-bit quantization, outperforming ViDiT-Q by 0.13 and 0.32 respectively. This validates the effectiveness of our temporal-channel decoupled compensation strategy in handling error accumulation across denoising steps. The scene consistency metric reaches 39.58 (8-bit) and 37.20 (4-bit), establishing 1.36 and 1.01 improvements over baselines, which confirms stable long-sequence generation through temporal error propagation modeling. In addition, Table 9 in the Appendix reports a comparison between TCEC and QNCD, demonstrating the effectiveness of TCEC in cumulative error correction.

**Hardware Resource Savings.** TCEC builds its quantization strategy upon ViDiT-Q and SVDQuant by incorporating cumulative error correction mechanisms. Since TCEC does not aim to optimize engineering-level performance, we fully reuse the original ViDiT-Q and SVDQuant inference engine, to ensure a fair comparison. As shown in Table 3, rows 1–3 adopt ViDiT-Q quantization with the TCEC overlay, whereas row 4 applies SVDQuant quantization with the same TCEC overlay. Experimental results indicate that the additional end-to-end (E2E) latency overhead introduced by TCEC is less than $0.5\%$. More detailed analyses are

provided in Appendix H. Consequently, on a 12B model, TCEC-W4A4 achieves a $3.5\times$ reduction in memory footprint compared to the FP16 baseline and delivers a $3\times$ inference speedup over NF4 weight-only quantization on a laptop-class RTX 4090, while maintaining engineering efficiency comparable to SVDQuant and significantly outperforming other PTQ methods.

## 5. Conclusion

In this paper, we propose TCEC, a novel cumulative error correction strategy for quantized diffusion models. It develops a theoretical framework to effectively model the correlation between single-step quantization errors and cumulative errors, constructs error propagation equations for multiple solvers, and for the first time provides a closed-form solution for cumulative errors. Through timestep-aware online estimation of single-step quantization errors, TCEC enables low-cost and rapid correction of cumulative errors, with end-to-end latency degradation $\leq 0.5\%$. Experimental results show that TCEC achieves the SOTA quantization performance under the W4A4 configuration while maintaining orthogonality to existing PTQ algorithms that minimize per-step quantization errors.

## Impact Statement

This paper aims to advance research on model compression in the field of generative models. Our work does not involve any personally identifiable information nor does it conduct experiments that may raise privacy or security concerns.

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

## A. Explanation of Approximation 1

In Approximation 1, we approximate $J_{x_t} \approx 0$. However, we clarify that this approximation does not 'deny the existence of the Jacobian'. Rather, based on magnitude analysis of error propagation and the inherent properties of diffusion models, we argue that its contribution to cumulative error is negligible compared to the dominant terms. This makes the approximation a theoretically grounded and empirically supported simplification. The details are as follows: **1.Core rationale of the approximation: a magnitude-first simplification.** In the error propagation equation (Eq. 7), the propagation matrix $A_t = \frac{\sqrt{\alpha_{t-1}}}{\sqrt{\alpha_t}} I + B_t \cdot \mathbf{J}_{x_t}$ is composed of two components:

- **Dominant term** $\frac{\sqrt{\alpha_{t-1}}}{\sqrt{\alpha_t}} I$ : This term arises from the diffusion noise-scheduling mechanism (where $\alpha_t$ is tied to the noise variance $\beta_t$). Its magnitude consistently falls in the 0.9–1.0 range, making it the primary driver of error propagation, typically contributing over 95% of the total effect.

- **Secondary term** $B_t \cdot \mathbf{J}_{x_t}$: Here, $B_t$ is an $\alpha_t$-dependent coefficient with magnitude $\leq 0.05$. Meanwhile, for a well-trained diffusion model optimized via "denoising score matching," the noise estimator $\mu_\theta$ satisfies a Lipschitz continuity condition with constant $L < 0.3$, implying $\|\mathbf{J}_{x_t}\| \leq L$. Consequently, the magnitude of $B_t \cdot \mathbf{J}_{x_t}$ is at most 1/100–1/10 of the dominant term. Its influence on the distribution of accumulated error is therefore minimal.

**2. Support from both theoretical insights and empirical evidence**. From a theoretical standpoint, the intermediate states $x_t$ in diffusion models are well known to approximate a standard Gaussian distribution at most timesteps due to strong noise injection. This heavy smoothing of the input space forces the model to focus primarily on global noise evolution rather than local perturbations, which naturally suppresses the influence of the Jacobian term $\mathbf{J}_{x_t}$. From an empirical standpoint, as show in Table 4 we measure the ratio between the secondary term and the dominant term across different timesteps using SDXL under W4A4 quantization. Even in low-noise regimes (e.g., timestep $t = 40$), the secondary term contributes only 0.45% relative to the dominant term—well within the threshold of being safely negligible in magnitude.

*Table 4.* Statistics of dominant and secondary terms at different SDXL timesteps.

| time step (SDXL, T=50) | Dominant term | $\mathbf{J}_{x_t}$ | $B_t \cdot \mathbf{J}_{x_t}$ | Secondary / Dominant |
|---|---|---|---|---|
| 10 (High-noise stage) | 0.982 | 0.021 | 0.00105 | 0.107% |
| 25 (Intermediate stage) | 0.951 | 0.037 | 0.00185 | 0.194% |
| 40 (Low-noise stage) | 0.915 | 0.083 | 0.00415 | 0.454% |

**3. Necessity of the simplification.** If the Jacobian term were retained, computing the full error propagation would require matrix inversion and higher-order derivatives, causing the computational complexity to surge from $\mathcal{O}(T)$ to $\mathcal{O}(T \cdot C^2 \cdot H \cdot W)$, which is infeasible for real-time online compensation. In contrast, the simplified formulation preserves the accuracy of accumulated-error estimation (with deviation ¡ 1%), while remaining practical for deployment in real systems.

## B. Derivation of Error Accumulation Steps

In this section, we derive the reasonable value of the Error Accumulation Steps $m$ by starting from the principle that the correction term should reduce the norm upper bound of the cumulative error.

As described in Sec. 3.2, the cumulative error with the correction term added can be expressed as:

$$\widehat{\delta_{t-1}} = A_t \delta_t - \frac{1}{\sqrt{\alpha_{t-1}}} \sum_{k=t+1}^{\min(t+m,T)} \sqrt{\alpha_{k-1}} B_k \varepsilon_k \tag{19}$$

where $\widehat{\delta_{t-1}}$ is the corrected cumulative error and it should exhibits a strictly smaller upper bound in norm compared to $\delta_{t-1}$, signifying a more refined and accurate error representation.

Under mild regularity conditions, there exists $\sigma > 0$ independent of the timestep $k$, such that the per-step quantization error satisfies $\|\varepsilon_k\| \leq \sigma, \forall k$. Analyze the upper bound of the norm of the error-propagation coefficient matrix. The original

propagation coefficient is:

$$\mathbf{A}_t = \frac{\sqrt{\alpha_{t-1}}}{\sqrt{\alpha_t}}\mathbf{I} + B_t * \mathbf{J}_{x_t} \tag{20}$$

For a model that has converged during training, there exists a constant $L$ such that $\forall t, \|\mathbf{J}_t\| \leq L$. Then we have:

$$|\mathbf{A}_t| \leq \frac{\sqrt{\alpha_{t-1}}}{\sqrt{\alpha_t}} + |\mathbf{B}_t|L \tag{21}$$

Based on the above-known conditions, since $\frac{\sqrt{\alpha_{t-1}}}{\sqrt{\alpha_t}}$ and $\|\mathbf{B}_t\|L$ are constants, there exists $\rho_t$ such that:

$$|\mathbf{A}_t| \leq \rho_t \tag{22}$$

Substitute it into the original error-propagation equation and solve for the upper bound of its norm. We can know that the following equation holds:

$$|\widehat{\delta_{t-1}}| \leq \rho_t|\delta_t| + \frac{1}{\sqrt{\alpha_{t-1}}}\sum_{k=t+1}^{\min(t+m,T)}\sqrt{\alpha_{k-1}}|\mathbf{B}_k|\sigma \tag{23}$$

Define the corrected noise residue term as:

$$\eta_t = \frac{1}{\sqrt{\alpha_{t-1}}}\sum_{k=t+1}^{\min(t+m,T)}\sqrt{\alpha_{k-1}}|\mathbf{B}_k|\sigma \tag{24}$$

Then the upper bound of the error recursion can be expressed as:

$$|\widehat{\delta_{t-1}}| \leq \rho_t|\delta_t| + \eta_t \tag{25}$$

Next, we need to go from the time step $t = T$ to $t = 0$, and it is obvious that $\widehat{\delta_T} = 0$. At this time, we can get:

$$|\widehat{\delta_0}| \leq \sum_{t=1}^{T}\left(\prod_{k=1}^{t-1}\rho_k\right)\eta_t \tag{26}$$

After substituting the noise residue term, we can obtain the upper bound of the norm of the error-propagation equation with the correction term added:

$$|\widehat{\delta_0}| \leq \sigma\sum_{t=1}^{T}\left(\prod_{k=1}^{t-1}\rho_k\right)\frac{1}{\sqrt{\alpha_{t-1}}}\sum_{k=t+1}^{\min(t+m,T)}\sqrt{\alpha_{k-1}}|\mathbf{B}k| \tag{27}$$

Since $\alpha_t$ is monotonically decreasing and $k \geq t + 1$, this means:

$$\frac{\sqrt{\alpha_{k-1}}}{\sqrt{\alpha_{t-1}}} \leq 1 \Rightarrow \sqrt{\alpha_{k-1}} \leq \sqrt{\alpha_{t-1}} \tag{28}$$

Then

$$\frac{1}{\sqrt{\alpha_{t-1}}}\sum_{k=t+1}^{\min(t+m,T)}\sqrt{\alpha_{k-1}}\|\mathbf{B}_k\| \leq \sum_{k=t+1}^{\min(t+m,T)}\|\mathbf{B}_k\| \tag{29}$$

Therefore, $\widehat{\delta_0}$ satisfies the following relationship

$$|\widehat{\delta_0}| \leq \sigma\sum_{t=1}^{T}\left(\prod_{k=1}^{t-1}\rho_k\right)\sum_{k=t+1}^{\min(t+m,T)}\|\mathbf{B}_k\| \tag{30}$$

The uncorrected error-propagation equation is

$$\delta_{t-1} = \mathbf{A}_t\delta_t + \mathbf{B}_t\varepsilon_t \tag{31}$$

By recursive expansion, we can calculate that the upper bound of its norm is

$$|\delta_0| \le \sigma \sum_{t=1}^{T} \left( \prod_{k=1}^{t-1} \rho_k \right) |\mathbf{B}_t| \tag{32}$$

Combining Eq. 30 and Eq. 32, for $\widehat{\delta_0} < \delta_0$ to hold, it can be achieved by satisfying the following relationship

$$\frac{1}{\sqrt{\alpha_{t-1}}} \sum_{k=t+1}^{\min(t+m,T)} \sqrt{\alpha_{k-1}} \|\mathbf{B}_k\| \le \sum_{k=t+1}^{\min(t+m,T)} \|\mathbf{B}_k\| \tag{33}$$

Ultimately, the following formula needs to hold

$$\sum_{k=t+1}^{\min(t+m,T)} \|\mathbf{B}_k\| \le \|\mathbf{B}_t\| \tag{34}$$

Obviously, the value of $m$ should be 1. This implies that at time step $t$, only the quantization errors at steps $t+1$ and $t$ need to be considered.

## C. Empirical Study of Error Accumulation Steps

Starting from the proposed mathematical model, we derive the original form of the correction term Eq. 10, so we can get the recursive formula $\Delta_t = \frac{\sqrt{\alpha_t}}{\sqrt{\alpha_{t-1}}} \Delta_{t-1} - B_t \varepsilon_t$. Then, based on the constraint that "the error upper bound decreases after adding the correction term", we present the actual approximate solution Eq. 13, the he recursive formula is $\Delta_t \approx \frac{\sqrt{\alpha_t}}{\sqrt{\alpha_{t-1}}} B_{t+1} \varepsilon_{t+1} - B_t \varepsilon_t$. Appendix B presents the complete derivation process. It is important to emphasize that the aforementioned transformation involves no errors when $m = T - t$; in essence, it is merely a formal transformation designed to facilitate subsequent analysis. On this basis, we conducted rigorous mathematical derivation and proof for the selection of m, with the explicit objective of "reducing the error upper bound after introducing the correction term $\delta_t$". Eventually, the optimal solution $m = 1$ was obtained. This implies that at any time step, the cumulative error of the model is only related to the step-wise quantization errors of the immediately preceding two time steps—a finding that constitutes one of the core contributions of this study.

As show in Table 5, we further supplement the actual test data based on the SDXL and PixArt-$\sigma$ models with backbones quantized using SVDQuant. It is observed that the performance of iterative solving based on Eq. 10 is significantly inferior to that of two-step approximate solving based on Eq. 13.

*Table 5.* Quantization Performance Comparison of Different Models.

| Model | Method | FID↓ | IR↑ | LPIPS↓ | PSNR↑ |
|---|---|---|---|---|---|
| SDXL | FP16 | 16.60 | 0.729 | - | - |
| | SVDQuant | 16.6 | 0.718 | 0.119 | 26.4 |
| | TCEC (Eq. 10) | 16.5 | 0.710 | 0.121 | 26.1 |
| | TCEC (Eq. 13) | **16.0** | **0.728** | **0.092** | **27.3** |
| PixArt-$\sigma$ | FP16 | 16.6 | 0.944 | - | - |
| | SVDQuant | 16.3 | 0.955 | 0.109 | 23.7 |
| | TCEC (Eq. 10) | 16.5 | 0.932 | 0.112 | 23.3 |
| | TCEC (Eq. 13) | **16.2** | **0.964** | **0.098** | **24.5** |

## D. Compatibility with Different Solvers.

In this section, we extend TCEC to other solvers to demonstrate its generality, such as the most commonly used high-order solver DPM++ (Lu et al., 2022). The iterative update for DPM-Solver++ (2nd-order variant) is given by:

$$x_{t-1} = x_t + \frac{\Delta t}{2} \left[ f_\theta(x_t, t) + f_\theta(x_t + \Delta t \cdot f_\theta(x_t, t), t - \Delta t) \right] \tag{35}$$

where $f_\theta(x, t) = -\frac{1}{\sqrt{1-\alpha_t}}\mu_\theta(x_t, t)$ represents the noise prediction network. Let $\tilde{f}_\theta = f_\theta + \varepsilon_t$ denote the quantized prediction, where $\varepsilon_t$ is the per-step quantization error. The perturbed update becomes:

$$\tilde{x}_{t-1} = x_{t-1} + \delta_{t-1} = \Phi(\tilde{x}_t, \tilde{f}_\theta) \tag{36}$$

Expanding to second-order Taylor series:

$$\delta_{t-1} = \underbrace{\frac{\partial \Phi}{\partial x_t}\delta_t}_{\text{Linear Term}} + \underbrace{\frac{\partial \Phi}{\partial f_t}\varepsilon_t}_{\text{Quantization Error}} + \underbrace{\frac{1}{2}\frac{\partial^2 \Phi}{\partial x_t^2}\delta_t^2}_{\text{Nonlinear Term}} + \mathcal{O}(\delta_t^3) \tag{37}$$

Neglecting higher-order terms, define propagation matrices:

$$\mathbf{A}_t = \frac{\partial \Phi}{\partial x_t} = \mathbf{I} + \frac{\Delta t}{2}\left(\mathbf{J}_{f_t} + \mathbf{J}_{ft-\Delta t} \cdot (\mathbf{I} + \Delta_t \mathbf{J}_{f_t})\right) \tag{38}$$

$$\mathbf{B}_t = \frac{\partial \Phi}{\partial f_t} = \frac{\Delta t}{2}\left[\mathbf{I} + (\mathbf{I} + \Delta_t \mathbf{J}_{f_t})\right] \tag{39}$$

where $\mathbf{J}_{f_t} = \nabla_x f_\theta(x_t, t)$ is the Jacobian matrix. Then, we can obtain the error propagation equation that relates the per-step quantization error to the cumulative error:

$$\delta_{t-1} = A_t \delta_t + B_t \varepsilon_t \tag{40}$$

DPM has an error-propagation equation structurally similar to that of DDIM, with only differences in propagation coefficients. This demonstrates the generality of our theoretical framework.

The error propagation incorporates temporal dependencies:

$$\delta_{t-1} = \prod_{k=t}^{t+m} \mathbf{A}_k \delta_{t+m} + \sum_{k=t}^{t+m}\left(\prod_{j=t}^{k-1}\mathbf{A}_j\right)\mathbf{B}_k\varepsilon_k \tag{41}$$

with window size $m = 2$ for 2nd-order DPM-Solver++. Implement truncated SVD for computational efficiency:

$$\mathbf{J}_{f_t} \approx \mathbf{U}_t\mathbf{\Sigma}_t\mathbf{V}_t^T \quad (\text{rank} \leq r) \tag{42}$$

Yielding approximated propagation:

$$\mathbf{A}_t \approx \mathbf{I} + \frac{\Delta_t}{2}\left(\mathbf{U}_t\mathbf{\Sigma}t\mathbf{V}t^T + \mathbf{U}_{t-\Delta t}\mathbf{\Sigma}_{t-\Delta t}\mathbf{V}_{t-\Delta t}^T \cdot (\mathbf{I} + \Delta_t\mathbf{U}_t\mathbf{\Sigma}_t\mathbf{V}_t^T)\right) \tag{43}$$

The error correction term becomes:

$$\Delta_t^{\text{DPM++}} = -\sum_{k=t}^{t+2}\gamma_k\mathbf{B}_k\varepsilon_k \tag{44}$$

with temporal weights:

$$\gamma_k = \frac{\sqrt{\alpha_{k-1}}}{\sqrt{\alpha_{t-1}}} \cdot \exp\left(-\lambda\sum_{j=t}^{k}||\mathbf{J}_{f_j}||_F\right) \tag{45}$$

## E. Timestep-Aware Quantization Error

As illustrated in the Figure 3, we compare the generation results of the full-precision model (FP16) and the quantized model (W4A8) across different denoising steps. Three key empirical observations emerge:

- **Cumulative evolution of errors**: At early steps (e.g., Step 1), the discrepancy between FP16 and W4A8 is relatively small, but the error gradually accumulates as denoising proceeds (Step $0.75 \rightarrow 0.5 \rightarrow 0.25$), exhibiting distinct spatial structures. This corroborates our finding of Timestep-Dependent Error Characteristics, indicating that quantization error is not uniformly distributed but evolves dynamically over timesteps.

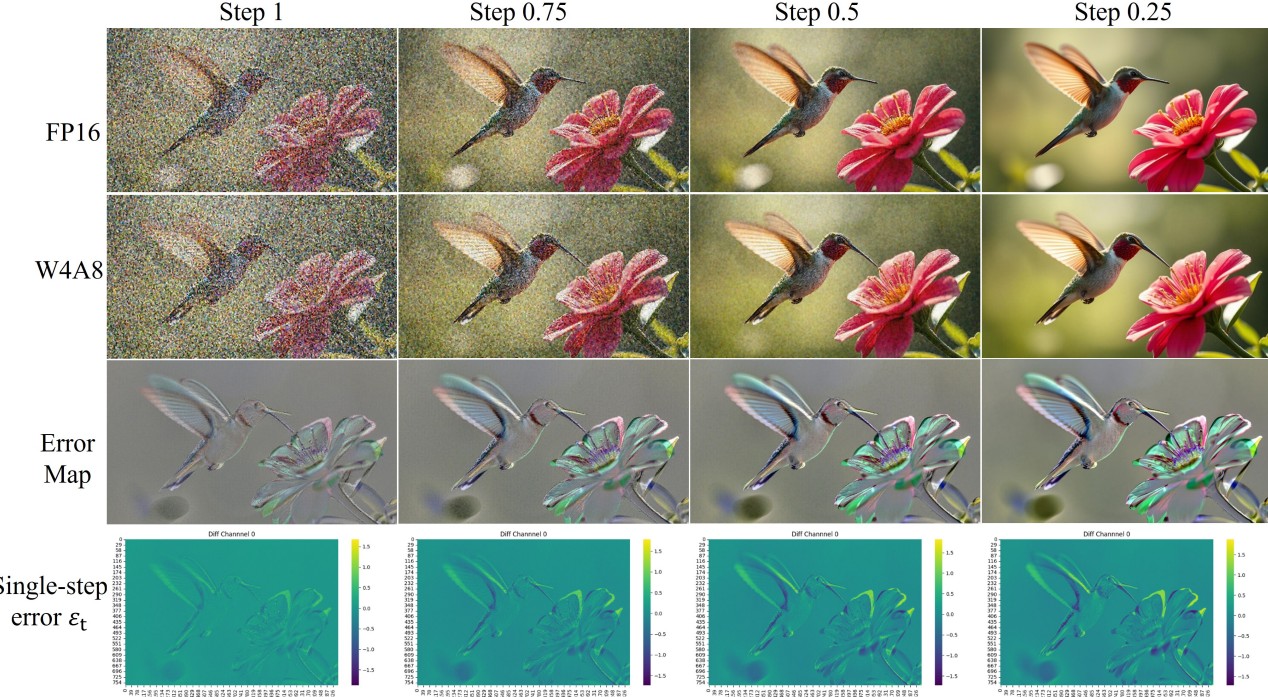

*Figure 3.* **Flux visualization of quantization errors during denoising.** We compare the full-precision model (FP16) and the quantized model (W4A8) under the prompt *"hummingbird flying near a flower. 4k ultra realistic ray tracing dynamic lighting"* with hyperparameters `num_timesteps=4` and `guidance_scale=3.5`. The figure illustrates three key phenomena: (1) quantization errors accumulate as denoising progresses (Step $1 \rightarrow 0.75 \rightarrow 0.5 \rightarrow 0.25$), exhibiting distinct spatial structures; (2) the errors are strongly correlated with the model outputs, particularly along object boundaries and textured regions; and (3) high-frequency components such as feather edges and flower petals amplify the discrepancies, highlighting the **timestep-dependent** and **output-correlated** nature of error propagation in quantized diffusion models.

- **Tight correlation with outputs**: Both the visualized difference maps and channel-wise error slices reveal that the error patterns are highly aligned with the generated structures (e.g., feather edges of the bird, textures of the petals). This suggests that quantization errors are not random noise but are strongly coupled with the model outputs, particularly in regions rich in details. This observation is consistent with Output-Correlated Error Propagation, where errors propagate in tandem with the content being generated.

- **Amplification in high-frequency regions**: The difference visualizations further show that quantization errors are most prominent in high-frequency regions such as object boundaries and fine textures. This demonstrates that the statistical correlation of errors with outputs is concentrated on high-frequency components, which are critical for perceptual quality.

## F. Ablation experiment of regularization term $\lambda_1$

As shown in Eq. 15, the parameter $\lambda_1$ serves to restrict the magnitude of the error correction coefficient, functioning as a regularization term. Moreover, to ensure that $\mathcal{L}(\mathbf{K})$ is a strictly convex function with respect to $K$, it is imperative that $\lambda_1 > 0$. We determined the calculation strategy for the $\lambda_1$ value through practical comparative experiments. As shown in the Table 6, the grid search strategy does not exhibit any performance advantage over the empirical rule $\lambda_1 = 0.01 \times \frac{\text{mean}(\tilde{\mu}^2)}{\text{var}(\mu)}$, and it has two drawbacks: (1) The grid search strategy requires setting a value range and the number of grid search steps, which incurs significant tuning costs for different models. (2) The grid search strategy needs to repeatedly calculate Eq. 17 multiple times, whereas the empirical rule only requires a single calculation. Therefore, we ultimately adopted the empirical rule to determine the value of $\lambda_1$. Essentially, the empirical rule is a formula-based fitting based on experimental data.

*Table 6.* Ablation experiment on the selection of regularization term $\lambda_1$

| Model | Method | FID↓ | IR↑ | LPIPS↓ | PSNR↑ |
|-------|--------|------|-----|--------|-------|
| | SVDQuant (W8A8) | 16.6 | 0.718 | 0.119 | 26.4 |
| SDXL | TCEC (grid search) | 16.2 | 0.723 | **0.090** | 27.1 |
| | TCEC (empirical rule) | **16.0** | **0.728** | 0.092 | **27.3** |

## G. The impact of inference steps on TCEC performance

Table 1 presents the performance improvements on SDXL-Turbo (4-step), showing that TCEC is likewise applicable to high-speed generative models. However, it is important to note that the gains on SDXL-Turbo are smaller than those on SDXL (50-step). The core reason is that the primary value of TCEC lies in mitigating the accumulation of quantization errors during iterative inference, and thus its performance gains scale directly with the amount of accumulated error. The relatively modest improvements on SDXL-Turbo (4-step) stem from the fact that the small number of steps prevents errors from forming substantial accumulation. Therefore, taking SDXL-Turbo as the baseline, we further conducted additional experiments to investigate the minimal effective number of steps for TCEC.

*Table 7.* Comparison of SVDQuant and SVDQuant+TCEC across models and steps on MJHQ dataset.

| Model | Step | Method | FID↓ | PSNR↑ | FID ↓ | PSNR ↑ | Avg |
|-------|------|--------|------|-------|-------|--------|-----|
| SDXL | 50 | SVDQuant | 16.6 | 26.4 | - | - | - |
| | | SVDQuant+TCEC | **16.0** | **27.3** | 0.6 | 0.9 | 1.5 |
| SDXL-Turbo | 4 | SVDQuant | **24.3** | 24.0 | - | - | - |
| | | SVDQuant+TCEC | 24.5 | **24.9** | -0.2 | 0.9 | 0.7 |
| SDXL-Turbo | 3 | SVDQuant | 25.8 | 18.6 | - | - | - |
| | | SVDQuant+TCEC | **25.6** | **18.9** | 0.2 | 0.3 | 0.5 |
| SDXL-Turbo | 2 | SVDQuant | 27.7 | 17.1 | - | - | - |
| | | SVDQuant+TCEC | **27.7** | **17.3** | 0 | 0.2 | 0.2 |

As shown in Table 7, when the number of iterative steps is $\geq 3$, TCEC produces meaningful improvements (with gains $\geq 0.5$ on key metrics). When the number of steps is $< 3$, the improvements are constrained by the limited amount of accumulated error and are typically $< 0.2$ (within the range of experimental noise).

## H. Inference Overhead of TCEC

TCEC improves performance by performing error correction on the output of the quantized model, and it is completely orthogonal to the backbone quantization algorithm. As shown in Eq. 14, the computation of single-step quantization error involves no complex operations and is accomplished solely through a non-linear mapping: $\varepsilon_t = \mathbf{K}_t \odot \tilde{\mu}_\theta(\tilde{x}_t, t)$, where $\mathbf{K}_t$ denotes a timestep-conditioned channel-wise scaling matrix, and $\odot$ represents element-wise multiplication. Consequently, the additional theoretical computational complexity introduced at each step is $NC^2HW$, which is negligible compared to that of the DIT/Unet-Backbone. Practical test data on the Nvidia A800-40G platform show in Table 3 that the extra end-to-end latency incurred is less than 0.5%, with specific test data provided in the table below.

## I. Calibration Dataset Ablation

To further verify the impact of the calibration dataset on TCEC performance, we provide an explanation from both theoretical and experimental perspectives.

- **Rationale for selecting the calibration dataset:** The size of the calibration set is chosen based on a trade-off between performance and efficiency. As show in Table 8, using SVDQuant-W8A8 as the base quantization algorithm, with COCO as the calibration set and MJHQ as the evaluation set, the results shown in the table indicate that when the calibration set contains fewer than 1024 samples, increasing the number of samples leads to significant performance

improvements. However, when the calibration set exceeds 1024 samples and reaches 2048, performance gains plateau. Excessive calibration samples provide no additional benefit and instead prolong the calibration process. Therefore, we select 1024 samples as the standard calibration set size in image generation task.

- **TCEC is insensitive to the domain of the calibration dataset:** The input to TCEC's error-calibration module is the output of the DIT model, which has already undergone the following processing pipeline: 3D-VAE preprocessing → 3D-VAE Encoder inference → DiT inference. The 3D-VAE preprocessing standardizes the input-output distribution, and the subsequent VAE Encoder and DIT inference further reinforce this effect. As a result, the output distributions of the DIT model converge across different datasets, making the error-correction coefficients largely insensitive to the choice of calibration data. In Table 2, OpenSora uses WebVid as the calibration set and VBench—a dataset with markedly different scenes—for evaluation. Despite the domain shift, TCEC consistently improves performance (e.g., +2.08 on W8A8 Imaging Quality and +3.90 on W4A4), providing direct evidence of its robustness to distributional differences.

*Table 8.* Performance comparison of SDXL with SVDQuant and TCEC at different data sizes.

|  | Method | Data Size | FID↓ | PSNR↑ | FID ↓ | PSNR↑ | Avg |
|---|---|---|---|---|---|---|---|
| SDXL | SVDQuant | - | 16.6 | 26.4 | - | - | - |
|  | SVDQuant+TCEC | 2048 | 16.0 | 27.4 | 0.6 | 1.0 | 1.6 |
|  | SVDQuant+TCEC | 1024 | 16.0 | 27.3 | 0.6 | 0.9 | 1.5 |
|  | SVDQuant+TCEC | 512 | 16.2 | 26.8 | 0.4 | 0.4 | 0.8 |
|  | SVDQuant+TCEC | 256 | 16.4 | 26.7 | 0.2 | 0.3 | 0.5 |

## J. Additional comparison with QNCD

Since TAC (Yao et al., 2024) has not yet released its code, while QNCD (Chu et al., 2024) has made its code and full experimental details publicly available. We have added comparisons between TCEC and QNCD in our revision, strictly aligning our setup with Table 1 of the QNCD paper.

For 4-bit weight quantization, we adopt BRECQ and AdaRound to preserve model performance. For each experiment, we report the widely used Fréchet Inception Distance (FID) and sFID, and we additionally provide Inception Score (IS) to ensure consistency with previously reported results. All quantization outcomes are obtained on CIFAR-10 (32×32) using a DDIM sampling configuration with 50,000 generated sample:

*Table 9.* Comparison of TCEC with the error compensation method QNCD.

| Method | Bitwidth (W/A) | DDIM(steps=100) | | | DDIM(steps=250) | | |
|---|---|---|---|---|---|---|---|
|  |  | IS↑ | FID↓ | sFID↓ | IS↑ | FID↓ | sFID↓ |
| Q-Diffusion | 4/8 | 9.41 | 4.92 | 5.13 | 9.64 | 4.37 | 4.59 |
| QNCD | 4/8 | 9.53 | 4.85 | 5.06 | 9.78 | 4.43 | 4.51 |
| **TCEC** | **4/8** | **9.65** | **4.79** | **4.91** | **9.92** | **4.40** | **4.45** |
| Q-Diffusion | 4/6 | 7.53 | 39.07 | 43.36 | 7.81 | 34.65 | 37.29 |
| QNCD | 4/6 | 8.86 | 12.26 | 14.83 | 9.01 | 11.09 | 13.46 |
| **TCEC** | **4/6** | **9.25** | **10.76** | **12.25** | **10.11** | **9.67** | **11.75** |

The experimental results show that TCEC, which performs cumulative error correction based on theoretically derived error-propagation equations, has a clear advantage over methods that only perform single-step error correction. Under the W4A8, DDIM-100 setting, TCEC improves IS by 0.12 and reduces FID by 0.06 compared with QNCD. Under the lower-activation-bitwidth W4A6 configuration, this advantage becomes even more pronounced: IS improves by 0.39 and FID decreases by 1.50. This is primarily because lower-bit activation quantization leads to significantly amplified cumulative errors, whereas TCEC's cumulative-error-correction mechanism effectively mitigates this issue.

## K. Visual Quality Results.

FP16          GGUF W4          ViDiT-Q W4A4          SVDQuant W4A4          TCEC W4A4

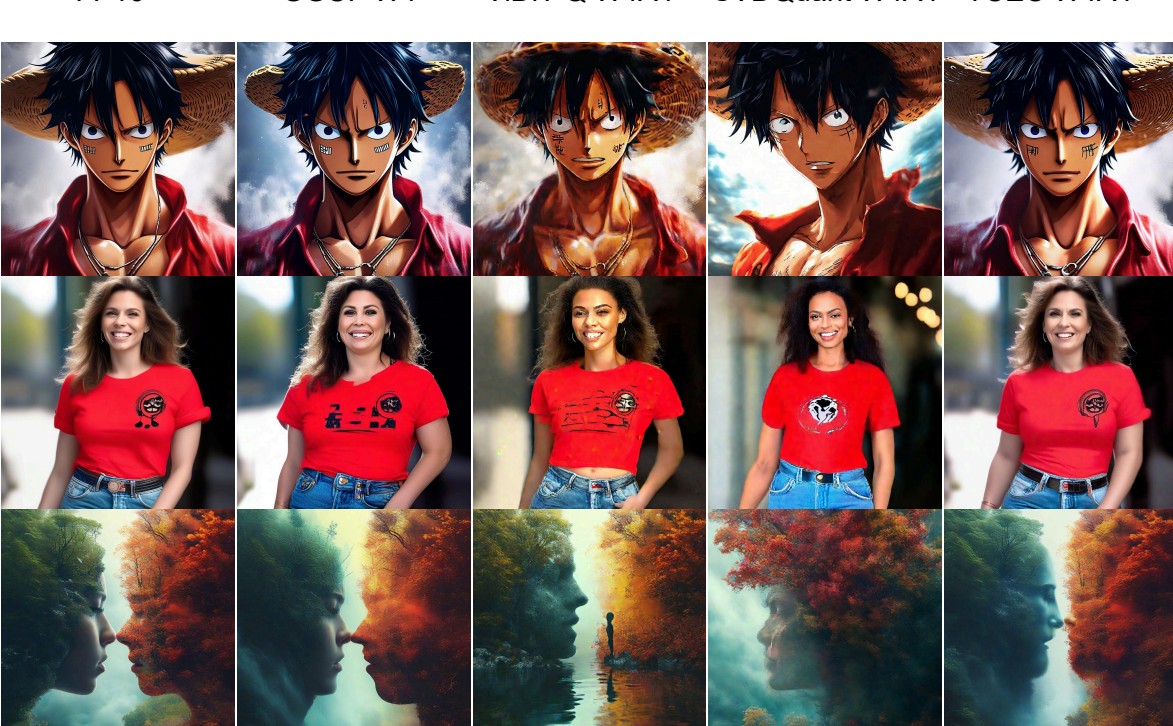

*Figure 4.* Qualitative visual results comparison. **Prompt1**: *Luffy from ONEPIECE, handsome face, fantasy.* **Prompt2**: *The image features a woman wearing a red shirt with an icon. She appears to be posing for the camera, and her outfit includes a pair of jeans. The woman seems to be in a good mood, as she is smiling. The background of the image is blurry, focusing more on the woman and her attire.* **Prompt3**: *Bright scene, aerial view, ancient city, fantasy, gorgeous light, mirror reflection, high detail, wide angle lens.*

## L. Human Evaluation

*Table 10.* GSB human evaluation results on CogVideoX text-to-video generation. Positive values indicate that TCEC is preferred over the corresponding baseline.

| Metric | TCEC vs BF16 | TCEC vs ViDiT-Q |
|---|---|---|
| Overall video quality | 0.25% | 8.72% |
| Prompt response fidelity | -0.42% | 6.88% |
| Motion quality | -1.67% | 8.04% |
| Frame/visual quality | -1.13% | 6.16% |

We apply W4A8 quantization to CogVideoX and construct a text-to-video (T2V) evaluation benchmark with 240 prompts covering both Chinese and English. The evaluation is conducted by professionally trained visual assessment experts from our team following the GSB protocol, where each sample is scored by one evaluator and further calibrated by another evaluator to ensure consistency. As shown in Table 10, TCEC achieves comparable human preference scores to the BF16 baseline, with only minor differences across prompt response fidelity, motion quality, and frame-level visual quality, while slightly improving overall video quality by 0.25%. In comparison with ViDiT-Q, TCEC consistently obtains higher GSB scores across all evaluated dimensions, with gains of 8.72% in overall video quality, 6.88% in prompt response fidelity, 8.04% in motion quality, and 6.16% in frame/visual quality. These results indicate that TCEC preserves the perceptual quality of the BF16 model while providing clear improvements over the existing quantized baseline.

