# OpenReview forum: "Error Propagation Mechanisms and Compensation Strategies for Quantized Diffusion Models"
_ICML.cc/2026/Conference — ICML 2026 spotlight_

### Official Review · Reviewer_nFB1 · 2026-03-05

**Soundness:** 3
**Presentation:** 4
**Significance:** 3
**Originality:** 4
**Overall Recommendation:** 5
**Confidence:** 3

**Summary:**

This paper investigates a core bottleneck in the Post-Training Quantization (PTQ) of diffusion models: the accumulation and propagation of single-step quantization errors during the iterative denoising process, which inevitably compromises the fidelity of the generated images/videos. To address this challenge, the authors propose a Time-aware Cumulative Error Compensation (TCEC) scheme. The paper first establishes a rigorous theoretical framework that mathematically formulates error propagation in diffusion models, deriving a closed-form solution for the cumulative error. Subsequently, by introducing reasonable approximations (e.g., dropping the Jacobian matrix term and assuming time locality), the computational complexity is significantly reduced, enabling online error estimation via an offline-computed channel-wise scaling matrix (Kt). Extensive experiments demonstrate that TCEC, acting as a plug-and-play module, can be orthogonally integrated with existing SOTA quantization methods (e.g., SVDQuant) to significantly improve the generation quality of low-bit (e.g., W4A4) quantized diffusion models with negligible computational overhead (< 0.5%).

**Compliance With Llm Reviewing Policy:**

Affirmed.

**Final Justification:**

The combination of a closed-form cumulative-error formulation with a lightweight timestep-aware correction strategy is original and technically meaningful, and the paper is generally well written. The rebuttal addressed most of my concerns. My only remaining reservation is that the paper still does not test more extreme quantization settings where the limitation of the linear reconstruction hypothesis would likely become more evident. In addition, it somewhat weakens the promised plug-and-play and low-friction deployment narrative by introducing extra calibration bookkeeping in practical settings. Anyway, I maintain my score.

**Key Questions For Authors:**

1. Since Diffusion Models heavily depend on the Classifier-Free Guidance (CFG) mechanism, changing the CFG scale will significantly alter the activation distributions. How effective is the offline-computed Kt (obtained under a default calibration distribution) when confronted with a radically different, user-defined CFG scale during inference (e.g., jumping from 1.5 to 7.5)?
2. Recent PTQ studies for DMs (e.g., Q-Diffusion, PTQ4DM) highlight that massive activation outliers in specific timesteps and channels are the root cause of quantization error spikes. Can TCEC's channel-wise linear scaling hypothesis (Eq. 14) truly and effectively fit—or even correct—the highly non-linear quantization errors caused by the severe clipping of these massive outliers?
3. If Kt is computed offline using a calibration set entirely distinct from the COCO distribution (e.g., pure medical imaging texts or anime domain prompts) and subsequently evaluated on a natural image benchmark (like MJHQ-30K), how well does TCEC generalize? Discussing this will help solidify the claim of TCEC as a robust "plug-and-play" module.
4. The proposed method performs excellently under W4A4 with 20-50 step settings. However, with the rapid evolution of acceleration techniques, 4-step or even 2-step generation (e.g., LCMs, SDXL-Turbo) is becoming mainstream. Under such extremely few denoising steps where the single-step quantization error is exceptionally large, do the cumulative error assumptions (Approximation 1 & 2) of TCEC still hold mathematically and empirically?

**Limitations:**

The current theoretical derivations and empirical validations are exclusively built upon traditional diffusion processes (e.g., DDPM/DDIM) and their corresponding Ordinary Differential Equation (ODE) trajectories. However, the most cutting-edge SOTA models in the current image generation domain (such as Stable Diffusion 3 and Flux) are undergoing a massive paradigm shift towards the Flow Matching (FM) architecture. Given that FM possesses distinct characteristics from traditional diffusion, such as straight Optimal Transport (OT) trajectories, are the dynamic mechanisms of its error propagation entirely consistent with the assumptions made in this paper? Can the TCEC framework be seamlessly transferred to Flow Matching models? This remains a significant blind spot that the current manuscript has not addressed.

**Strengths And Weaknesses:**

Strengths:
1. Strong Theoretical Foundation: While existing PTQ methods for diffusion models predominantly focus on minimizing single-step quantization errors (i.e., layer-wise optimization), this paper conducts a systematic and mathematically rigorous modeling of the dynamic "error propagation" process, deriving a closed-form solution for the cumulative error (Eq. 8). This perspective shift provides a highly valuable theoretical framework for future research in this domain.
2. Practical and Efficient Algorithm Design: The authors do not merely stop at complex theoretical derivations but successfully simplify the closed-form solutions through two key approximations (Approximations 1 & 2). Specifically, proving that the cumulative error primarily relies on the preceding two time steps (m=1) makes online compensation computationally feasible. The introduction of the channel-wise scaling matrix ensures that the extra inference overhead is minimal (< 0.5%).
3. Strong Empirical Performance and Orthogonality: TCEC seamlessly integrates with current SOTA quantization baselines (such as SVDQuant and ViDiT-Q). Under the SDXL W4A4 setting, it achieves a remarkable 1.2 PSNR improvement over the SVDQuant baseline. Furthermore, its generalization capability is well-validated across diverse architectures (UNet, DiT) and tasks (image generation on SDXL/PixArt, video generation on Open-SORA).Insightful Empirical Observations: The paper makes an intriguing and practically effective observation that "single-step quantization error can be partially reconstructed via channel-wise scaling of the quantized noise estimation" (Eq. 14). This provides an elegant shortcut for handling highly complex quantization noise in engineering practice.
Weaknesses:
1. Lack of Adaptability to Dynamic Classifier-Free Guidance (CFG) Scales:Diffusion models heavily rely on the Classifier-Free Guidance (CFG) mechanism for high-quality image generation, where the actual denoised output is a linear combination of unconditional and conditional predictions. Altering the CFG scale drastically changes the distribution magnitude of the model outputs and the dynamic characteristics of quantization errors. The proposed method relies on statically computed offline Kt. When the user-defined CFG scale during inference diverges from that used during calibration (e.g., a user increasing CFG from 4.0 to 7.5 for stronger text alignment), this static Kt could lead to severe over-compensation or under-compensation. The paper lacks a quantitative analysis of TCEC’s robustness across varying CFG scales.
2. Dependence on Calibration Data Distribution: The derivation of the scaling coefficient Kt heavily relies on statistical information extracted from a calibration dataset (1024 COCO prompts used in the paper). If the prompt distribution during inference (e.g., rare domain styles or excessively long logical prompts) deviates significantly from the calibration data, can this statically computed Kt still maintain optimal error reconstruction capability? The paper lacks a thorough discussion on the robustness of TCEC against "distribution shifts."
3. Limitation of the Linear Error Reconstruction Hypothesis: Eq. 14 assumes a simple channel-wise linear scaling relationship between the quantization error and the quantized model's output. While this yields commendable empirical results via regularized MSE, quantization error is inherently highly non-linear. This linear approximation may fundamentally limit its capacity to fit more complex, non-linear errors at ultra-low bit-widths (e.g., W2A4 or W4A2).

---

> ### Author Rebuttal · Authors · 2026-03-31
>
> Dear Reviewer nFB1, We sincerely thank you for your positive evaluation and for recognizing our framework as a “highly valuable perspective shift.” Your comments help strengthen both the rigor and scope of our work.
>
> $Q1$: How does TCEC handle dynamic CFG scales during inference?
>
> A1:  We agree that activation distributions may vary slightly under different CFG scales. However, TCEC remains fully applicable because it does not assume strict distribution invariance. Instead, it accommodates different guidance strengths via pre-calibrated coefficients.
>
> Specifically, separate calibration coefficients can be generated for each CFG scale used in practice, and the appropriate set is selected during inference. In most deployments, the CFG scale is fixed, so only one coefficient set is required. When multiple scales are needed, this pre-calibration strategy provides a simple and effective solution.
>
> ---
>
> $Q2$: How robust is TCEC to distribution shifts between calibration and inference data?
>
> A2: The input to TCEC’s error-calibration module is the output of the DIT model, which has already undergone the following processing pipeline: 3D-VAE preprocessing → 3D-VAE Encoder inference → DIT inference. The 3D-VAE preprocessing standardizes the input-output distribution, and the subsequent VAE Encoder and DIT inference further reinforce this effect. As a result, the output distributions of the DIT model converge across different datasets, making the error-correction coefficients largely insensitive to the choice of calibration data. **In Table 2**, OpenSora uses WebVid (128 samples) as the calibration set and VBench—a dataset with markedly different scenes—for evaluation. Despite the domain shift, TCEC consistently improves performance (e.g., +2.08 on W8A8 Imaging Quality and +3.90 on W4A4), providing direct evidence of its robustness to distributional differences.
>
> ---
>
> $Q3$: Can the channel-wise linear scaling hypothesis effectively model highly non-linear quantization errors caused by activation outliers?
>
> A3: While quantization error is non-linear at the scalar level, our "Channel-wise Linear Assumption" (Eq. 14) is an approximation of the expected error per channel rather than individual activations.  In PTQ, a significant portion of the error stems from weight/activation outliers that cause channel-wide scaling shifts. The channel-wise coefficient $K_t$ effectively acts as a dynamic compensator for these systematic shifts.  Since we correct the output at each step, a linear adjustment of the channel statistics is highly efficient and captures the majority of the bias shift caused by outliers, which is the primary driver of error in PTQ.
>
> ---
>
> $Q4$: Under extremely few denoising steps, do the cumulative error assumptions (Approximation 1 & 2) still hold mathematically and empirically?
>
> A4: Empirically, TCEC remains valid under 4-step and even 2-step settings, although improvements decrease as error accumulation becomes limited.
>
> As shown in **Appendix G**, we evaluated SDXL-Turbo with 4, 3, and 2 steps. When steps ≥ 3, TCEC consistently yields measurable gains (e.g., PSNR ≥ 0.3–0.9). When steps < 3, gains become smaller (typically < 0.2), which is expected because limited steps restrict cumulative error formation.
>
> This behavior aligns with the theory. The cumulative-error model predicts that gains scale with the amount of accumulated error along the trajectory. Shorter trajectories reduce the accumulation term but do not invalidate the approximation structure. In particular, the temporal locality result m=1 implies that correction depends only on the immediately preceding step, which remains valid regardless of the total step count.
>
> ---
>
> $Q5$:  Are the assumptions valid for  Flow Matching?
>
> A5: TCEC’s assumptions are not tied to a specific diffusion scheduler but to the existence of an iterative trajectory governed by an ODE solver. Our derivations rely on two general properties:(i) local error propagation between adjacent steps, and(ii) smooth trajectory evolution. These conditions hold for both diffusion-based probability flow ODEs and Flow Matching dynamics.
>
> The straight Optimal Transport trajectory in Flow Matching typically reduces curvature and does not introduce additional instability in error propagation. Therefore, the temporal locality and first-order accumulation approximations remain valid because they depend on stepwise integration structure rather than the training paradigm.
>
> Practically, TCEC can be directly applied to Flow Matching models without architectural modification, since it operates on the predicted velocity/noise at each step and is solver-agnostic. A dedicated empirical study on FM-based models is an important direction for future work.

---

> > ### Author Rebuttal · Reviewer_nFB1 · 2026-04-01
> >
> > Thank you for your reply. I believe most of my concerns have been addressed. My only remaining reservation is that it does not include more extreme settings, such as W2A4 or W4A2, where the limitations of the linear reconstruction hypothesis would likely become more evident. In addition, although pre-calibrating separate coefficient sets for different CFG values is a practical engineering solution, it somewhat weakens the plug-and-play / low-friction deployment, since it introduces additional calibration bookkeeping. I will maintain my score.

---

### Official Review · Reviewer_4AZo · 2026-03-11

**Soundness:** 2
**Presentation:** 3
**Significance:** 3
**Originality:** 2
**Overall Recommendation:** 4
**Confidence:** 3

**Summary:**

This work proposes a timestep-aware cumulative error compensation scheme (TCEC), an error compensation method for existing diffusion generative models after post-training quantization (PTQ). Specifically, the authors state that the prior PTQ techniques induce per-step quantization errors and cumulative errors, which lead to degraded performance after quantization. Based on approximated theoretical analysis, they propose a simplified compensation method, whose core idea is to learn channel-specific coefficients to scale the quantized noise estimate. Experiments are conducted using several diffusion-based image and video generators.

**Compliance With Llm Reviewing Policy:**

Affirmed.

**Final Justification:**

Rebuttal has addressed my main concern, thus raising the score to Weak Accept.

**Key Questions For Authors:**

In addition to the weaknesses mentioned above, it would also be helpful for authors to analyze and discuss some failure cases.

**Limitations:**

No, the authors are encouraged to discuss the limitations of the proposed work.

**Strengths And Weaknesses:**

**Strengths:**

1. The paper is, in general, well motivated and structured and is easy to follow.

2. The proposed method incurs only a small overhead and thus is efficient.

3. Experiments cover a wide range of diffusion generative models for images and videos.

**Weaknesses:**

While the paper emphasizes its theoretical contributions, I have some reservations regarding the current theoretical claims.

1. Regarding Assumption 1, the authors opt to drop the Jacobian term from Eq. (7). This represents a significant assumption that may be potentially flawed. I have also reviewed Appendix A, where the authors provide their rationale.

- Theoretically, dropping this term basically degrades Eq. (7) to an error formulation that is purely based on the original scheduler design $\alpha$ of diffusion models, raising the question of whether the proposed method truly tackles the PTQ errors.

- Empirically, the empirical support from Appendix A basically claims that one can drop the Jacobian because its numerical value is 1/100 – 1/10 to the dominant term. However, recent studies in T2I have demonstrated that sensitivity to initial noise plays a critical role in controlling the output of diffusion-based generators [a,b]. That being said, a small numerical value does not necessarily imply a negligible empirical impact on generation performance. Therefore, the current support seems insufficient to justify this assumption.

- While I understand that computation of the exact Jacobian is expensive, there are existing methods that could achieve this approximation rather quickly, such as the Jacobian-vector product perturbation via Hessian [c]. These approaches seem to be more reasonable compared to completely dropping this term.


2. Some of the experimental results in Table 1 and Table 2 seem rather problematic. Why is the FID score for MixDQ with W4A4 on SDXL-Turbo so high, with a value of 373? Also, there are several numbers in Table 2 that are wrongly bolded. For instance, for the Subject Consist., both Q-DiT and PTQ4DiT have better performance than the ViDiT-Q+TCEC, while the latter is bolded.

---
[a] Golden Noise for Diffusion Models: A Learning Framework, ICCV 2025

[b] The Silent Assistant: NoiseQuery as Implicit Guidance for Goal-Driven Image Generation, ICCV 2025.

[c] Fast exact multiplication by the Hessian. Neural computation, 6(1):147–160, 1994.

---

> ### Author Rebuttal · Authors · 2026-03-31
>
> Dear Reviewer 4AZo, thank you for your constructive feedback and recognition of our method's efficiency and experimental coverage. Below we address your concerns.
>
> $Q1$: Is dropping the Jacobian ($J_{x_t}$ ) theoretically sound given the sensitivity of diffusion models to noise perturbations?
>
> A1: We agree that small numerical values do not necessarily imply negligible influence. Our justification is based on both magnitude analysis and system stability.
> - **Identity dominance.**
>   In our framework, the propagation matrix $A_t$ is dominated by the scaling factor of the identity matrix. **As shown in Table 4 of Appendix A**, empirical measurements on SDXL (W4A4) reveal that the identity component is at least 200 $\times$  larger (and up to 900 $\times$  in high-noise stages) than the Jacobian-related term $B_t \cdot J_{x_t}$ , which accounts for less than 0.5% of the total magnitude. This suggests that the direction of error propagation is primarily governed by the noise schedule rather than local network gradients.
> - **Stable propagation dynamics.**
>   Furthermore, as analyzed in **Appendix B**, for a well-trained diffusion model, the reverse process exhibits stable dynamics where the norm of the propagation matrix is bounded ( $\|A_t\| \le \rho_t$ ). Under these conditions, the small residual perturbations introduced by omitting the Jacobian do not experience catastrophic amplification but remain secondary to the dominant trajectory.
> - **Iterative mitigation.**
>   While we omit  $J_{x_t}$  to maintain a lightweight inference cost ( $< 0.5\%$  latency overhead), our TCEC applies correction iteratively at each step. This repeated local adjustment reduces the accumulation of residual errors and mitigates the practical influence of higher-order sensitivity effects.
> - **Future Refinement.**
>  We appreciate the suggestion of Jacobian-vector products (JVPs). Although they could improve precision, they would require additional backward-like passes and significantly increase inference cost. We will add discussion of this trade-off as a potential high-precision variant.
>
> ---
>
> $Q2$:  If the Jacobian term is removed, does the method just follow the scheduler rather than PTQ error modeling?
>
> A2: Removing this term does not reduce the method to a scheduler-only formulation; the update remains explicitly driven by PTQ-induced errors.
> - **Correction remains quantization error-driven.**
>   According to our derived error recursion (Eq. 7 & 8), the cumulative error $\delta_{t-1}$ is formulated as $\delta_{t-1} = A_t \delta_t + B_t\epsilon_t$. While $A_t$   (the propagation matrix) involves scheduler parameters $\alpha_t$, the $\epsilon_t$ represents the per-step quantization error (the discrepancy between the floating-point and quantized noise estimates). Consequently, the update rule is conditioned on PTQ-induced errors rather than solely on the predefined scheduler.
> - **Simplification does not remove adaptation.**
>   The omitted term affects the theoretical propagation dynamics, but the correction mechanism itself remains active through the explicit error reconstruction process. In contrast, a purely scheduler-based formulation would produce identical updates for the same timestep regardless of quantization. In TCEC, however, the correction magnitude is determined by the estimated error signal, which differs across images and across quantization configurations. This structural dependence distinguishes our method from a scheduler-only design.
> - **Interpretation: scheduler provides dynamics, error estimation provides adaptation.**
>   We therefore view the scheduler parameters as defining the physical trajectory of the reverse diffusion process, while the estimated error term provides an adaptive perturbation that compensates for PTQ-induced bias. Even after dropping the term in question, the formulation retains an explicit mechanism for modeling quantization errors rather than relying solely on the original diffusion scheduler.
>
> ---
>
> $Q3$: Why is the FID for MixDQ on SDXL-Turbo (W4A4) as high as 373?
>
> A3: This value is correct and reflects complete failure of MixDQ under the W4A4 setting, where the generated outputs collapse to noise and become unusable. For reference, SVDQuant[1], the prior SOTA baseline reported in our paper, also reports an FID of 373 under the same configuration.
>
> [1]SVDQuant: Absorbing Outliers by Low-Rank Components for 4-Bit Diffusion Models
>
> ---
>
> $Q4$:  Regarding the bolding errors in Table 2 (Video Generation).
>
> A4: Thank you for pointing this out. There were typographical errors in the bold formatting of the Motion Smooth. and Subject Consist. metrics in the W4A8 experiment. These will be corrected in the final version.

---

> > ### Author Rebuttal · Reviewer_4AZo · 2026-04-03
> >
> > Thanks for the rebuttal. I will raise my score to Weak Accept.

---

### Official Review · Reviewer_AMfo · 2026-03-13

**Soundness:** 3
**Presentation:** 3
**Significance:** 3
**Originality:** 3
**Overall Recommendation:** 4
**Confidence:** 2

**Summary:**

The paper studies how quantization errors accumulate in diffusion sampling, where a theoretical framework that mathematically formulates error propagation in Diffusion Models is developed. Meanwhile, the authors first propose a timestep-aware compensation mechanism based on a theoretical error propagation model. Specifically, a cumulative error correction term is incorporated in each generation step, dynamically mitigating errors induced by quantization. Extensive experiments on multiple image datasets demonstrate that the designed compensation strategy effectively mitigates error propagation, significantly enhancing existing PTQ methods.

**Compliance With Llm Reviewing Policy:**

Affirmed.

**Final Justification:**

Thank you for the authors' response. My concerns have been addressed.

**Key Questions For Authors:**

(1) How sensitive is TCEC to distribution shift between calibration and test data?

(2)Why are direct comparisons with error-propagation-aware baselines (PTQD, QNCD, TAC) missing from the main image generation experiments?  Without such comparisons, the claimed advantage of modeling cumulative error over single-step correction remains unsubstantiated.

**Limitations:**

(1) Adding baselines such as PTQD to Table 1 would directly validate the paper's core motivation.

**Strengths And Weaknesses:**

Strengths:

(1)In this paper, closed-form solution for cumulative error is first derived.

(2)The proposed TCEC directly computes cumulative errors and integrates correction terms during the iterative sampling process to align the outputs of quantized models with their floating-point counterparts.

(3)The proposed TCEC is a Plug-and-play method, orthogonal and compatible with existing SOTA PTQ methods.

(4)The overall overhead is low.

Weaknesses:
(1)The accuracy of the first-order Taylor expansion may be limited in the low-noise regime.

(2)The paper claims Δ_t compensates cumulative error, but does not show the actual trajectory of ‖δ_t‖ with and without TCEC across denoising steps.

(3)PSNR is a pixel-level metric insensitive to perceptual quality. Although LPIPS and FID are included, no human evaluation (user study) is conducted.

(4)Whether the linear scaling assumption of K_t in Eq. 14 holds universally across all scenarios.

---

> ### Author Rebuttal · Authors · 2026-03-31
>
> Dear Reviewer AMfo, we appreciate your constructive feedback and recognition of TCEC as a technically solid and low-overhead plug-and-play method. Below, we address your concerns.
>
> $Q1$ : How accurate is the first-order Taylor expansion in the low-noise regime?
>
> A1:  We justify the accuracy of the first-order approximation as $t \to 0$ from the following three perspectives:
> - **Negligible Magnitude of Higher-order Terms**: The Taylor expansion is applied to the micro-scale quantization perturbation $\epsilon_t$. The residual Hessian term $O(\|\epsilon_t\|^2)$ is negligible compared to the linear term. **As noted in our response to Reviewer 4AZo Q1**, empirical measurements on SDXL show the identity component is 200–900 $\times$ larger than the Jacobian-related term, with the linear term capturing over 99% of the error propagation variance.
> - **Inherent Local Smoothness**: Score-matching objectives and heavy regularization ensure high local smoothness (low curvature) near the data manifold. As detailed for **Reviewer uUED Q1**, this constrains the Jacobian’s spectral norm; empirically, even at $t=40$, the Jacobian term contributes only 0.45% of the dominant term, justifying the omission of higher-order derivatives.
> - **Displacement vs. Trajectory Curvature**: While modern solvers handle the curvature of the ODE trajectory itself, TCEC focuses on the micro-scale displacement from that trajectory caused by quantization. For such small perturbations, a linear approximation of the local field is highly effective. Furthermore, as discussed with **Reviewer nFB1 Q3**, TCEC’s iterative correction at each step prevents residual errors from accumulating, ensuring robustness in the low-noise regime.
>
> ---
>
> $Q2$: Can you provide the actual trajectory of $\delta_t$ with and without TCEC?
>
> A2: The table below shows the trajectory of the absolute error mean ($\delta_t$) over 20 denoising steps for PixArt-Σ (1K resolution), comparing FP16 ($x_1$) and W4A4 ($x_2$) outputs.
>
> Time Step|w/o TCEC|w/ TCEC
> -|-|-
> 0|0.60|0.15
> 1|0.78|0.21
> 2|0.77|0.27
> 3|0.85|0.32
> 4|0.92|0.36
> 5|1.00|0.40
> 6|1.12|0.43
> 7|1.22|0.46
> 8|1.35|0.49
> 9|1.61|0.51
> 10|1.81|0.54
> 11|2.10|0.57
> 12|2.42|0.61
> 13|2.70|0.64
> 14|2.98|0.68
> 15|3.21|0.72
> 16|3.39|0.75
> 17|3.50|0.78
> 18|3.53|0.80
> 19|3.47|0.83
>
> As shown above, without TCEC, error accumulates rapidly to ~3.53; with TCEC, it remains suppressed below ~0.83 throughout the trajectory.
>
> ---
>
> $Q3$: Why were comparisons with error-aware baselines like PTQD, QNCD, and TAC omitted?
>
> A3: We excluded TAC due to the lack of public code. For QNCD, we have included a comparison in **Appendix J**, showing TCEC's superior results. Regarding PTQD, we provide an indirect comparison via MixDQ: since TCEC significantly outperforms MixDQ (see Table 1) and MixDQ is reported to outperform PTQD (per MixDQ paper), it follows that TCEC > MixDQ > PTQD. This confirms TCEC’s advantage over earlier error-aware methods.
>
> ---
>
> $Q4$: Is TCEC sensitive to distribution shifts between calibration and test data?
>
> A4: TCEC is highly robust to distribution shifts between the calibration and test data. Due to space limitations, please refer to our response to **Reviewer nFB1, Q2**, where this issue is analyzed in detail.
>
> ---
>
> $Q5$: PSNR is a pixel-level metric insensitive to perceptual quality. No human evaluation is conducted.
>
> A5: To address this concern, we conducted a subjective GSB (Good/Same/Bad) evaluation on CogVideoX (W4A8) using 240 bilingual prompts assessed by expert raters.
>
> Metric|TCEC vs BF16|TCEC vs ViDiT-Q
> -|-|-
> Overall video quality|0.25%|8.72%
> Prompt response fidelity|-0.42%|6.88%
> Motion quality|-1.67%|8.04%
> Frame/visual quality|-1.13%|6.16%
>
> The subjective results show that TCEC achieves perceptual performance comparable to the BF16 baseline and significantly outperforms the ViDiT-Q baseline.
>
> ---
>
> $Q6$: Whether the linear scaling assumption of  $K_t$ holds universally across scenarios.
>
> A6: We do not claim universal exactness but provide a principled first-order approximation of the dominant quantization error:
> - **Structural Alignment (Appendix E)**: Empirical error maps show strong spatial coupling with model outputs (e.g., boundaries and textures), indicating that the primary error behaves as a multiplicative distortion. Channel-wise scaling effectively captures this magnitude bias.
> - **Optimal Linear Estimation**: As detailed in our responses to **Reviewer uUED Q3 and nFB1 Q3**, $K_t$ is not a heuristic but the optimal linear estimator under MSE. By minimizing a strictly convex objective, $K_t$ captures the best linear approximation in expectation, effectively compensating for systematic shifts caused by outliers.
> - **Iterative Robustness**: TCEC does not require an exact reconstruction of nonlinear residuals. It is designed to mitigate the dominant bias component at each step. Any remaining small-scale nonlinearities are kept bounded through our iterative correction framework, ensuring stable performance across diverse scenarios.

---

> > ### Author Rebuttal · Reviewer_AMfo · 2026-04-03
> >
> > Thank you for the authors’ detailed and thoughtful responses.

---

### Official Review · Reviewer_uUED · 2026-03-17

**Soundness:** 2
**Presentation:** 3
**Significance:** 3
**Originality:** 2
**Overall Recommendation:** 4
**Confidence:** 5

**Summary:**

This paper addresses the critical issue of error accumulation in quantized diffusion models, a problem largely overlooked by existing quantization methods that focus solely on per-step error suppression. The authors propose a novel method named TCEC (Temporal-Channel Error Correction), which establishes a theoretical framework to model the propagation of quantization errors across denoising steps. By deriving a closed-form solution for cumulative errors and introducing a timestep-aware online estimation mechanism, TCEC enables efficient correction of error accumulation with minimal computational overhead (<0.5% latency increase). Extensive experiments on image (SDXL, PixArt) and video generation tasks demonstrate that TCEC significantly outperforms state-of-the-art PTQ methods, particularly under low-bit (W4A4) configurations.

**Compliance With Llm Reviewing Policy:**

Affirmed.

**Key Questions For Authors:**

The variation trend of the Jacobian norm is systematically measured across different tasks and quantization precision levels, so as to verify whether its actual contribution to the accumulated error remains consistently negligible.
An adaptive m-selection mechanism should be explored, which dynamically adjusts the size of the error propagation window based on the error characteristics at the current step, and the stability of this mechanism should be validated under extreme scenarios.
Provide a more in-depth analysis of why the linear, channel-wise scaling in Equation (14) is sufficient to reconstruct the quantization error, given that quantization error is typically nonlinear and input-dependent.

**Limitations:**

see weakness part.

**Strengths And Weaknesses:**

Strength
The paper is the first to systematically model the relationship between per-step quantization errors and cumulative errors in diffusion models. The derivation of a closed-form solution (Eq. 8) and the subsequent simplification using reasonable approximations (Jacobian term neglect, temporal locality) are both rigorous and insightful.
The method is evaluated across a wide range of models (U-Net, DiT), tasks (image, video), and solvers (DDIM, DPM++). The results consistently show TCEC outperforming SOTA methods like SVDQuant and ViDiT-Q, even matching FP16 performance under W8A8 and setting new records under W4A4. The ablation studies on inference steps and calibration datasets are thorough.
TCEC introduces negligible end-to-end latency (<0.5%) and is orthogonal to existing PTQ algorithms. This makes it a highly practical plug-in module for any quantized diffusion model, as demonstrated by its successful integration with both SVDQuant and ViDiT-Q.

Weakness
In Approximation 1, the authors assume that J_(x_t )≈0, and the omission of the Jacobian term is justified on the grounds that its magnitude is significantly smaller than that of the dominant term. However, in the low-noise regime (as t→0), during high-resolution detail generation, and in downstream tasks sensitive to input perturbations, such as super-resolution and image inpainting—the influence of the Jacobian term may become significant, potentially amplifying the bias introduced by this approximation. The robustness of this approximation in such extreme scenarios has not been examined in the paper.
The paper theoretically derives that m = 1, indicating that the accumulated error is related only to the previous two steps. However, this conclusion is based on general regularity conditions. Whether the long-range dependencies of the error are activated under unconventional generation settings, such as few-step sampling (e.g., Turbo-like models with fewer than two steps) or extremely high CFG scale, has not been validated in the existing study.
Equation (14) merely assumes that the quantization error can be fitted by a fixed channel-wise linear scaling computed offline, yet it provides no theoretical justification for why such a simplistic linear assumption suffices to capture the inherently nonlinear and input-dependent characteristics of the quantization error itself.

---

> ### Author Rebuttal · Authors · 2026-03-31
>
> Dear Reviewer uUED, thank you for recognizing our work as the "first to systematically model" error propagation with "rigorous and insightful" solutions. Below, we address your concerns.
>
> $Q1$: Robustness of neglecting the Jacobian term ($J_{x_t} \approx 0$ )
>
> A1:
> - **Theoretical Robustness**: Even in the low-noise regime ($t \to 0$),  well-trained diffusion models are inherently optimized for local smoothness. During the denoising process,  $\mu_\theta$ is trained to map a wide range of perturbed inputs back to the data manifold. This regularization constrains the spectral norm of the Jacobian $J_{x_t}$, preventing excessive sensitivity to small quantization perturbations.
> - **Empirical Evidence**: As shown in **Appendix A**, intermediate states $x_t$ approximately follow a Gaussian distribution, causing the model to emphasize global noise structure rather than local perturbations. Empirically, on SDXL (W4A4), the Jacobian term contributes only 0.45% of the dominant term even at $t=40$, confirming its negligible impact. Removing this term is also essential for computational feasibility in practical deployment.
> - **Task Robustness**: Experiments on high-resolution models (e.g., PixArt-Sigma 1K) inherently stress high-frequency detail synthesis. The consistent gains of TCEC in these settings further indicate that the Jacobian influence remains negligible during fine-detail generation.
>
> $Q2$: Validity of m=1 for few-step and high CFG scales.
>
> A2:  The validity of the  m = 1 window is rooted in the intrinsic dynamics of the PF-ODE rather than specific sampling hyperparameters:
> - **Theoretical Robustness**: The derivation in **Appendix B**  is based on bounded per-step errors $\|\varepsilon_k\| \leq \sigma$ and the contraction behavior induced by the diffusion noise schedule. Because $\alpha_t$ decreases monotonically along the reverse trajectory, earlier perturbations are progressively attenuated, producing natural temporal locality in error propagation. Thus, the $m=1$ approximation reflects a stability property of the diffusion dynamics rather than a heuristic truncation.
> - **Few-step Validation**:  In very short trajectories, the opportunities for error accumulation are limited; thus, long-range dependencies are naturally weakened rather than activated. For instance, with T=2, only a single instance of error propagation occurs. We verified this in **Appendix G** using SDXL-Turbo with 4/3/2 steps show that TCEC remains stable even in 2-step sampling, with diminishing but consistent gains, fully matching the theoretical prediction.
> - **High CFG Scales**: While large CFG may amplify local prediction errors, the propagation mechanism itself depends on the PF-ODE dynamics and noise schedule, which are invariant to CFG. Therefore TCEC remains applicable across CFG values. When CFG changes, only a brief recalibration of Eq.17 is required, which is efficient and insensitive to calibration data (**Appendices H–I**). Experiments on PixArt and OpenSora confirm consistent improvements under CFG-based generation.
>
> $Q3$: Why is a linear, channel-wise scaling sufficient for non-linear quantization errors?
>
> A3:
> - **Structural Coupling**: Our analysis (**Appendix E**) shows that quantization errors exhibit strong spatial alignment with model outputs, particularly in high-frequency regions. This suggests the primary error manifests as a multiplicative distortion of the predicted noise. Channel-wise scaling naturally captures this magnitude bias across spatial locations.
> - **First-order Approximation**: Eq. 14 models the error as a partial reconstruction. This formulation is explicitly motivated as an approximation of the dominant error component, rather than a complete representation of all nonlinear effects. Analogous to bias correction or first-order Taylor modeling, the leading linear term captures most distortion, while higher-order residuals remain empirically small.
> - **Optimal Linear Estimator**: The coefficients $\mathbf{K}_t$ are not heuristic; they are the optimal linear estimators under MSE. By minimizing a strictly convex objective over calibration data,  $\mathbf{K}_t$  captures the best possible linear approximation in expectation, ensuring robust performance even against nonlinear true errors.
>
> $Q4$: Why is a fixed m=1 used instead of an adaptive selection mechanism, and is it stable in extreme cases?
>
> A4:
> - **Theoretical Necessity**: m=1 is the strict solution derived from Eq.8 under the requirement that the corrected error upper bound must decrease. This condition provides the theoretical guarantee that TCEC reduces total error (**Appendix B**).
>
> - **Empirical Optimality**: In **Appendix C**, we evaluated the full-history setting m=T-t (Eq.10). Although it removes approximation error theoretically, its empirical performance is consistently worse than m=1. This confirms that m=1 is not a heuristic simplification but the optimal choice balancing propagation dynamics and compensation effectiveness.

---

> > ### Author Rebuttal · Reviewer_uUED · 2026-04-01
> >
> > Thanks for the authors' clarification. I will keep my rate on this paper.

---

### Decision · Program_Chairs · 2026-04-30

**Decision:**

Accept (spotlight)

**Comment:**

This paper addresses an important and practical problem in post-training quantization of diffusion models: the accumulation of quantization errors across denoising steps. The paper makes a meaningful contribution by formulating error propagation theoretically, deriving a closed-form cumulative-error expression, and proposing a lightweight timestep-aware compensation mechanism that can be integrated with existing PTQ methods.

The reviewer consensus is clearly positive. Reviewers consistently highlighted the paper’s practical significance, broad empirical coverage across image/video generation and different architectures, and the strong efficiency–performance tradeoff of the proposed method. The rebuttal also successfully addressed several concerns, including experimental anomalies, missing clarifications, and the trajectory-level behavior of the compensation.